# Regulation of plant immunity through histone H3 β-hydroxybutyrylation-mediated transcriptional control

Qiutao Xu[1,9], Zhengting Chen[2,9], Rui Wang[3,9], Xuan Ma [4], Yuhang Duan[5], Du Hai[5], Jing Chen[6], Zhongyi Cheng [7], Lu Zheng [5], Junbin Huang[5], Jisen Zhang [1], Yu Zhao [8] ✉ & Xiaoyang Chen [6] ✉

Histone lysine β-hydroxybutyrylation (Kbhb) is a novel type of histone acylation whose prevalence and function remain unclear in plants. Here, we systematically characterized H3K9bhb in rice (*Oryza sativa*) and found that it is enriched at transcription start site (TSS) regions and serves as an active mark for gene expression. We demonstrated functional differences between H3K9bhb and the well-studied H3K9ac histone modification, particularly in regulating genes involved in the rice immune response. We also discovered that the exogenous application of β-hydroxybutyrate induces H3K9bhb deposition and promotes the expression of defense-related genes, enhancing disease resistance. Furthermore, we identified OsSRT1, OsSRT2, and OsHDA705 as key players in the removal of Kbhb. We also conducted a Kbhb proteomic analysis and identified 2159 Kbhb sites on 1128 proteins, providing a valuable resource for future studies of Kbhb. This research advances our understanding of plant epigenetic regulation by establishing Kbhb as a cellular post-translational modification with the potential to regulate plant immune responses.

Histone octamers, composed of two copies each of H2A, H2B, H3, and H4, are the main components of nucleosomes. Various covalent modifications on the N-terminal tails of histones affect gene transcription by regulating chromatin states, facilitating diverse biological processes in animals and plants, such as organ development, cellular differentiation, cellular metabolism, and the immune response[1–9]. Histone acetylation and methylation are the two best-characterized histone modifications. In addition, many histone lysine acylation modifications are mediated by short-chain fatty acids in animals and plants, including lysine crotonylation (Kcr), succinylation (Ksu), butylation (Kbu), lactylation (Kla), glutarylation (Kglu), benzoylation (Kbz), 2-hydroxyisobutyrylation

(Khib), and β-hydroxybutyrylation (Kbhb)[10–15]. The functions of these recently discovered acylation modifications in the regulation of gene expression are well understood in animal cells; however, their functions in plants have not been fully elucidated.

Protein post-translational modifications (PTMs), including phosphorylation and acetylation, play vital roles in the interactions between pathogens and plants[16–18]. Histone acetylation is one of the best-studied PTMs and is critical for modulating immune responses through the coordinated actions of histone acetyltransferases (HATs) and histone deacetylases (HDACs)[19]. For example, in Arabidopsis, the HAC1/5-containing complex interacts with NPR1 and TGA transcription

[1]State Key Laboratory for Conservation and Utilization of Subtropical Agro-bioresources, College of Agriculture, Guangxi University, Nanning, China. [2]State Key Laboratory of Rice Biology and Breeding, China National Rice Research Institute, Hangzhou, China. [3]College of Life Sciences, Yangtze University, Jingzhou, China. [4]College of Agriculture, Ningxia University, Yinchuan, China. [5]State Key Laboratory of Agricultural Microbiology, Huazhong Agricultural University, Wuhan, China. [6]Anhui Province Key Laboratory of Crop Integrated Pest Management, Anhui Agricultural University, Hefei, China. [7]Jingjie PTM BioLab Co. Ltd, Hangzhou, China. [8]National Key Laboratory of Crop Genetic Improvement, Huazhong Agricultural University, Wuhan, China. [9]These authors contributed equally: Qiutao Xu, Zhengting Chen, Rui Wang. ✉e-mail: zhaoyu@mail.hzau.edu.cn; cxy084@ahau.edu.cn

factors to promote H3 acetylation and the transcription of pathogenesis-related genes during salicylic acid-triggered immunity[20]. Conversely, in rice, HDACs such as HDT701 and SRT2 negatively regulate innate immunity by modifying H4 acetylation at defense gene promoters during pathogen infection[9,21]. Emerging evidence indicates that histone acylation also plays important roles in plant immunity. For example, lysine Khib and Ksu modifications are strongly associated with resistance to pathogens[7,22]. However, whether histone acylation regulates plant immunity remains to be fully elucidated.

Kbhb is a recently discovered PTM in which β-hydroxybutyrate (BHB) is enzymatically attached to exposed lysine residues on histones[10]. In animal histones, 46 histone Kbhb sites have been identified, with H3K9bhb being the best characterized to date[10,23]. Starvation induces the deposition of H3K9bhb at histones located at the promoters of mouse genes involved in starvation-responsive metabolic pathways, resulting in their upregulation[10]. Increased BHB availability enhances H3K9bhb levels on the promoters of Brain-Derived Neurotrophic Factor (BDNF) genes and upregulates their expression, reducing depression and depressive behaviors in mice[24]. Similarly, BHB also epigenetically modifies the H3K9 residues of *Foxo1* and *Ppargc1a*, leading to their upregulation and thereby influencing carbon flow in CD8[+] memory T cells[25]. Besides H3K9, Kbhb modifications are also observed on H3K4, H3K27, H3K56, H4K5, and H4K16[10,23]; however, their effect on chromatin state and gene expression requires further investigation. Kbhb are also found on non-histone proteins in mitochondria and the cytoplasm, suggesting that they have a wide variety of functions[23]. Though extensively studied in animal cells, the presence and roles of Kbhb in plants remain unknown.

In addition to their role in histone acetylation, HATs and HDACs have other acyltransferase and deacylase activities in animal and plant cells; for example, mammalian CBP/p300, a histone acetyltransferase, serves as a writer of histone Kbhb[23], while lysine acetyltransferase-2A (KAT2A) functions as a histone H3 succinyltransferase[11]. HDAC1 and HDAC2 act as histone Kbhb deacylases in animal cells[23]. Some plant HDACs also double as histone deacylases; for example, the rice (*Oryza sativa*) sirtuin histone deacetylase SRT2 is a histone decrotonylase[14]. In *Arabidopsis thaliana*, HDA6 and HDA9 are histone 2-hydroxyisobutyrylation erasers[26]. In rice, analogous roles are played by OsHDA705, OsHDA716, OsSRT1, and OsSRT2 in the de-2-hydroxyisobutyrylation[7]. However, the enzymes responsible for histone Kbhb modifications in plants remain unknown.

Rice, a staple food for over half the world's population, is crucial for global food security, but its yield is constrained by various pathogens[27]. Rice false smut, caused by the biotrophic fungal phytopathogen *Ustilaginoidea virens*, is a devastating grain disease in rice-growing regions worldwide[28]. In this study, we performed a proteome-wide screen for the Kbhb in rice and explored the role of H3K9bhb in disease resistance. We found *U. virens* infection increased H3K9bhb levels in rice, suggesting that this modification may play a pivotal role in the defense response. We supplemented rice plants with exogenous BHB, which enhanced H3K9bhb deposition on the defense-related genes and promoted their expression, conferring resistance against *U. virens*. Furthermore, we found that OsSRT1, OsSRT2, and OsHDA705 were responsible for the removal of histone Kbhb in rice. Our findings emphasize the robust nature of Kbhb as a histone mark regulated by plant HDACs and its pivotal role in plant–pathogen interactions. Our findings highlight Kbhb as an important protein PTM that might actively participate in various metabolic pathways.

## Results

### Identification and genome-wide profiling of histone H3K9bhb in rice

To investigate whether BHB-mediated histone Kbhb occurs in plants, we first analyzed the BHB content in the rice cells. The results showed that BHB was detected not only in the cytoplasm but also in the

nucleus (Supplementary Fig. 1a). Next, we conducted immunoblotting of rice, *Arabidopsis*, maize (*Zea mays*), and tobacco (*Nicotiana benthamiana*) histone proteins using specific antibodies (anti-Kbhb and anti-H3K9bhb) that were developed, characterized, and reported in a previous study[10] and here (Supplementary Fig. 2). In addition, we conducted immunoprecipitation using anti-H3K9bhb with two selected nuclear-localized proteins to further verify the specificity of H3K9bhb in rice cells (Supplementary Fig. 3). Our results revealed that Kbhb is a conserved chromatin mark in plants (Fig. 1a, b). To explore the genome-wide occupancy of histone Kbhb, we performed anti-H3K9bhb chromatin immunoprecipitation followed by high-throughput sequencing (ChIP-seq) in rice spikelets (Supplementary Fig. 4a). In total 40,053 peaks (23,296 marked genes; Supplementary Data 1) of H3K9bhb were identified. Most H3K9bhb peaks were located in genic regions (Fig. 1c). Among the H3K9bhb-marked genes, 89.2% were protein-coding (non-transposable element [TE]) genes (Fig. 1d), and most (66.9%) had one H3K9bhb peak (Fig. 1e, f). The H3K9bhb marks were highly enriched at transcription start sites (TSSs) of protein-coding genes in comparison to TE genes (Fig. 1g, left). Motif analysis of the promoters of these H3K9bhb-marked genes identified 53 motifs, comprising binding sites for well-characterized transcription factors, including those from the bZIP, NAC, WRKY, and MYB families (Supplementary Fig. 5). Genes longer than 1 kb showed a strong H3K9bhb enrichment in the TSS regions, while these marks were positioned at the gene body of shorter genes (≤1 kb, Fig. 1g, right). The ChIP-seq data were validated using a ChIP-qPCR analysis of three randomly selected H3K9bhb-enriched regions and two randomly selected regions not modified by H3K9bhb (Supplementary Fig. 6).

### Functional divergence of H3K9bhb and H3K9ac

Given that both H3K9bhb and H3K9ac occur at the same lysine residue and are associated with transcriptional activation, it is important to assess whether they function redundantly or exhibit distinct regulatory roles in gene expression. To investigate the correlation between H3K9bhb and H3K9ac, we conducted ChIP-seq using an H3K9ac-specific antibody (Supplementary Fig. 4a). Our analysis identified 14,499 genes (17,710 peaks) that showed co-occurrence of H3K9ac and H3K9bhb signals (Fig. 2a, b and Supplementary Data 2), which may arise from cell-to-cell heterogeneity. In addition, we identified 6983 genes specifically marked by H3K9ac alone and 8798 genes marked by H3K9bhb alone (Fig. 2a and Supplementary Data 2), suggesting distinct functional roles for these two histone modifications. Sequential chromatin immunoprecipitation (ChIP-reChIP) combined with qPCR revealed that H3K9ac and H3K9bhb co-occupy the same genomic loci in a mutually exclusive manner (Supplementary Fig. 7). A metaplot analysis showed that the genomic distribution profiles of both H3K9ac and H3K9bhb were generally similar across rice genes (Fig. 2b); however, genes uniquely marked by H3K9bhb exhibited histone mark enrichment within the gene body in addition to the TSS, in contrast with the more TSS-specific H3K9ac profile (Fig. 2b). To investigate the functional relationship between H3K9bhb and H3K9ac in the regulation of gene expression, we analyzed the expression levels and chromatin accessibility of genes modified by different combinations of these histone marks. Genes marked only by H3K9ac showed higher expression levels and more open chromatin states than those with only H3K9bhb or no mark at all (Fig. 2c). Notably, genes showing both H3K9ac and H3K9bhb occupancy (despite potential cell heterogeneity) exhibited the highest expression levels and most open chromatin states (Fig. 2c). This observation suggests that H3K9ac plays a dominant role in promoting gene expression, while H3K9bhb may serve as a complementary mark to enhance the effects of H3K9ac. Further analysis of the functional roles of H3K9ac and H3K9bhb using a GO analysis revealed that genes marked solely by H3K9ac were primarily enriched in functions involved in plant development and metabolic pathways (Fig. 2d), while genes specifically marked by

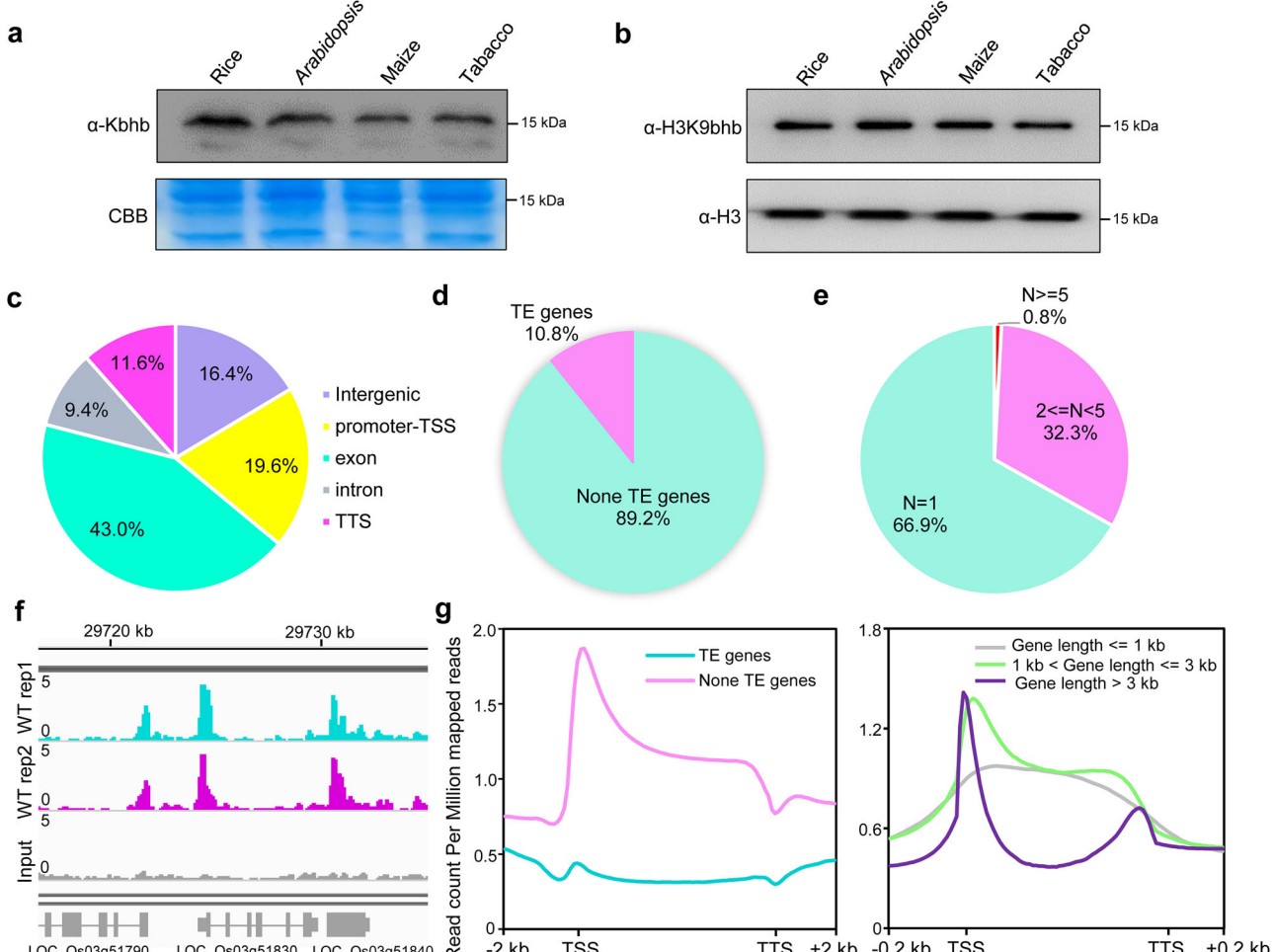

**Fig. 1 | Distribution of H3K9bhb in rice genome. a** Detection of histone lysine Kbhb in rice, *Arabidopsis*, maize, and tobacco by immunoblotting. Coomassie blue (CBB) staining was used as a loading control. Images shown are representative of two independent experiments. **b** Detection of H3K9Kbhb in rice, *Arabidopsis*, maize, and tobacco by immunoblotting. Histone H3 was used as a loading control. Images shown are representative of two independent experiments. **c** Genomic distribution of H3K9bhb peaks in the rice genome. **d** Pie chart of H3K9bhb marked TE and non-TE genes in rice. **e** Statistics on the number of H3K9bhb peaks per gene. **f** Integrative Genomics Viewer screenshot of H3K9bhb peaks. **g** Left: Metaplots of H3K9bhb in TE and non-TE genes. Right: Metaplots of H3K9bhb on genes of different lengths. TE: transposon elements. TSS: transcriptional start site. TTS: transcriptional terminal site.

H3K9bhb were enriched in responses to stress, especially biotic stimuli (defense response to fungus). These differences indicate the divergent roles of these histone marks in regulating gene function.

## Comparative analysis of H3K9bhb, histone H3 acetylation, and histone H3 methylation modifications

To further explore the epigenetic landscape shaped by H3K9bhb, we expanded our analysis to include well-characterized histone modifications such as H3 acetylation and methylation, aiming to uncover how these marks coordinate or compete in regulating gene expression. Using k-means clustering on chromatin modification heatmaps, we identified three distinct clusters of genes enriched in the various marks. Genes associated with H3K9bhb (Fig. 3a), which along with the positive marks H3K4ac, H3K9ac, and H3K4me3, were enriched in genes belonging to Clusters 1 and 3 (Fig. 3a). By contrast, H3K9me2 showed a contrasting pattern to the other euchromatin marks, being notably enriched in Cluster 2 genes (Fig. 3a). This observation suggests that H3K9bhb functions as an active chromatin modification, as it was positively correlated with the active histone marks H3K4ac, H3K9ac, and H3K4me3 (Fig. 3b), and was positively related to gene expression (Fig. 3c). In addition to several active histone marks, genes in Cluster 1 also showed a relatively high abundance of repressive H3K27me3 marks (Fig. 3a). These genes exhibited relatively low expression levels

and had highly open chromatin states (Fig. 3d). A gene ontology (GO) analysis revealed that genes in Cluster 1 were notably enriched in processes such as response to stimulus, response to oxidative stress, and response to stress, among others (Fig. 3e), while genes in Clusters 2 and 3 were more enriched in molecular and cellular processes, such as peptidyl-threonine phosphorylation, cell cycle regulation, protein transport, RNA splicing, and cell development (Supplementary Fig. 8). In addition, most of genes in Cluster 1 were highly induced by various abiotic and biotic stresses (Fig. 3f), indicating a potential role for H3K9bhb in regulating genes for stress-induced activation.

## Application of BHB in promoting H3K9bhb and disease-resistance

Short-chain acyl-CoAs are the substrates for many protein acylation reactions. In animal cells, β-hydroxybutyryl-CoA is generated from cellular BHB[10]. Our ChIP-seq data showed that H3K9bhb was enriched at the genes involved in the defense response to fungus (Fig. 2d); thus, we speculated that BHB may enhance disease resistance in rice. To investigate this possibility, we treated rice plants with an exogenous application of 100 μM BHB and exposed them to various pathogens. The BHB treatment enhanced resistance to rice blast, rice false smut, and sheath blight (Fig. 4a–f). Consistent with the rice results, BHB pretreatment induced fungal resistance in rapeseed (*Brassica napus*

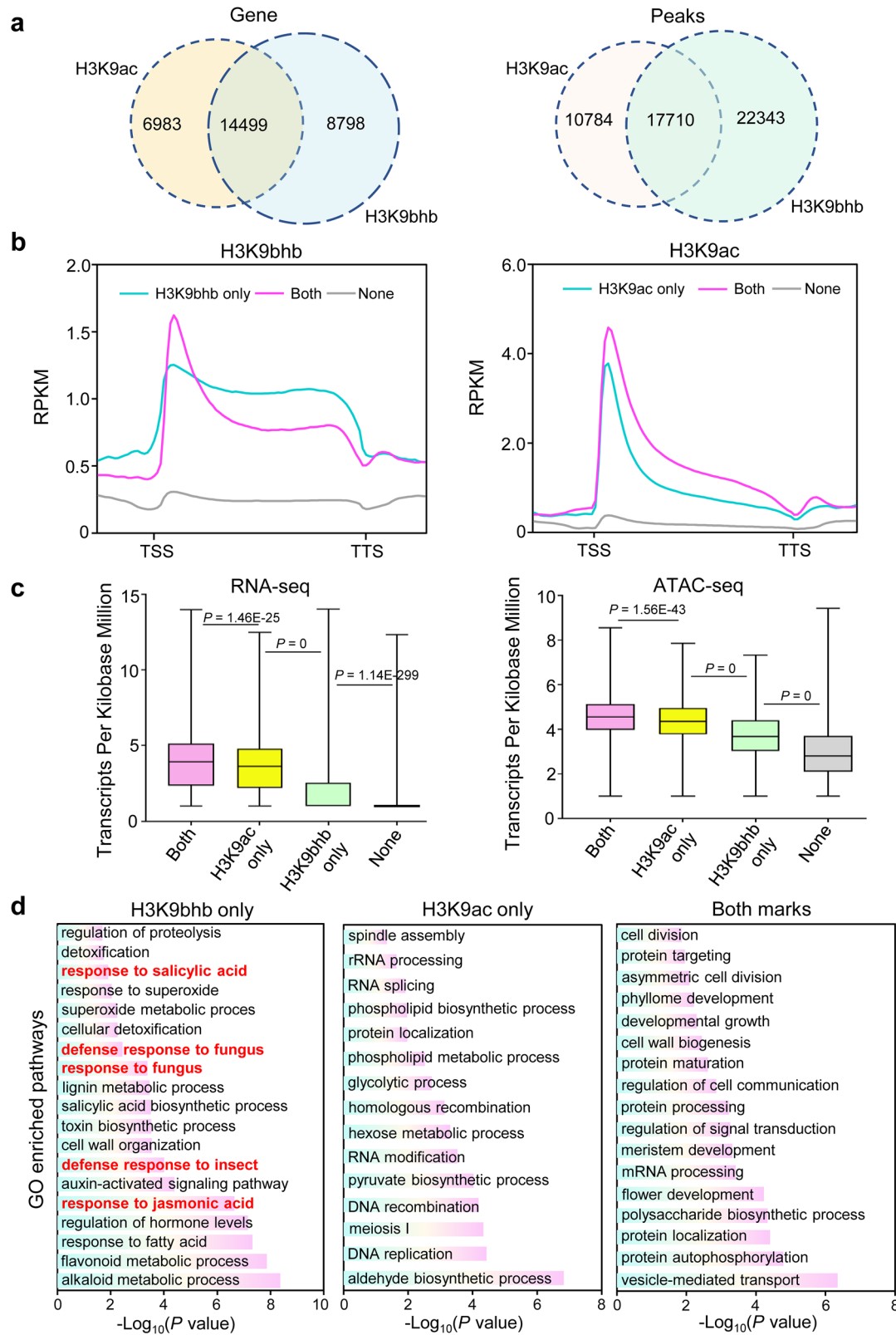

subsp. *napus*) and wheat (*Triticum aestivum*) (Supplementary Fig. 9). Treatment with 100 μM BHB activated rice immune responses, including the callose deposition, mitogen-activated protein kinase (MAPK) cascade, reactive oxygen species (ROS) burst, and the expression of defense-related genes (Fig. 4g–j). These findings suggest that BHB induces the rice immune response to enhance disease resistance.

To investigate whether exogenous BHB increases histone Kbhb levels in plants, we treated rice spikelets with various concentrations of BHB. Analysis of BHB content by liquid chromatography–tandem mass spectrometry (LC–MS/MS) confirmed an increase in BHB levels in rice cells (Supplementary Fig. 1b). Western blot analysis of treated rice spikelets revealed that histone Kbhb and H3K9bhb levels increased in a BHB dose-dependent manner, whereas a slight decrease in H3K9ac

**Fig. 2 | Comparative analysis between H3K9bhb and H3K9ac histone modifications. a** Left: Overlap of the H3K9ac and H3K9bhb marked genes. Right: Overlap of the H3K9ac and H3K9bhb peaks. **b** Metaplots of H3K9bhb and H3K9ac for different categories of genes in (**a**). RPKM: Reads Per Kilobase per Million mapped reads. TSS: transcriptional start site. TTS: transcriptional terminal site. None indicates genes without H3K9ac and H3K9bhb modifications. **c** Boxplots of gene expression levels (left) and ATAC signals (right) for H3K9bhb and H3K9ac across different categories of genes. For boxplots, horizontal lines show medians; box limits indicate the 25th and 75th percentiles. RNA-seq, RNA sequencing. ATAC-seq, assay for transposase accessible chromatin with high-throughput sequencing. None indicates genes without H3K9ac and H3K9bhb modifications. *P*-values were determined by a two-tailed, paired Student's *t*test. **d** Gene ontology (GO) enriched pathway analysis of genes specifically marked by H3K9bhb, H3K9ac, and those exhibiting both H3K9bhb and H3K9ac modifications as shown in (**a**). For panels **b** and **c**, different categories of genes with respect to H3K9bhb and H3K9ac modifications were based on the criteria presented in (**a**). None indicates genes without H3K9ac and H3K9bhb modifications.

levels was observed (Fig. 5a), suggesting that histone Kbhb levels are positively regulated by cellular BHB concentrations. To further test whether BHB-enhanced H3K9bhb levels promote the transcription of genes involved in disease resistance, we performed H3K9bhb ChIP-seq and RNA-seq (Supplementary Fig. 10) on control rice spikelets (CK) and rice spikelets treated with 100 μM BHB. ChIP-seq reads from BHB-treated and CK rice spikelets were plotted for the H3K9bhb target genes, revealing a significant genome-wide increase in this mark under the BHB treatment (Fig. 5b, c). These results were in line with the western blotting estimates of H3K9bhb levels in BHB-treated rice spikelets. A comparative study revealed that 6766 genes (> 1.5-fold, *FDR* < 0.05; Supplementary Data 3) had significantly increased H3K9bhb deposition in BHB-treated rice spikelets, whereas only 31 genes (>1.5-fold, *FDR* < 0.05) showed a significant decrease in H3K9bhb deposition compared with the CK treatment (Fig. 5d). A GO pathway enrichment analysis revealed that hyper-H3K9bhb genes were enriched in responses to fungus, jasmonic acid, and host-defense pathways (Fig. 5e). In parallel, an RNA-seq analysis revealed that 3898 genes were significantly upregulated (>2-fold, *P* < 0.05) and 3247 genes (> 2-fold, *P* < 0.05) were significantly downregulated in BHB-treated rice spikelets compared with CK spikelets (Fig. 5f and Supplementary Data 4). A GO pathway analysis revealed that the upregulated genes were enriched in similar pathways as were the genes with hyper-H3K9bhb levels in BHB-treated rice spikelets, particularly in the response to fungus (Fig. 5g). Furthermore, the upregulated genes also displayed a significant increase in H3K9bhb levels in BHB-treated rice spikelets (Fig. 5h). The correlation between dynamic changes in H3K9bhb and gene transcription in BHB-treated spikelets compared to CK spikelets showed a low positive correlation ($r = 0.11$; values between 0.1 and 0.3 indicate a low correlation, Fig. 5i). However, a relatively stronger correlation ($r = 0.42$; values between 0.3 and 0.7 indicate a moderate correlation) was observed within a subset of genes ($n = 1100$, Supplementary Fig. 11). Among these, 599 genes exhibited both upregulated expression and increased H3K9bhb levels in BHB-treated spikelets compared to CK spikelets (Fig. 5i and Supplementary Data 5). GO pathway enrichment analysis of the genes exhibiting both upregulated expression and hyper-H3K9bhb marks revealed their enrichment in defense response pathways (Fig. 5j). These data were further validated by ChIP-qPCR and RT-qPCR analyses (Supplementary Fig. 12). Since histone modifications can influence transcription by altering chromatin states, we tested this hypothesis by analyzing chromatin accessibility in BHB-treated (100 μM) and CK samples using assay for transposase-accessible chromatin with high-throughput sequencing (ATAC-seq) (Supplementary Fig. 13a). The results showed that BHB application enhanced chromatin accessibility (Supplementary Fig. 13b), and that genes with increased H3K9bhb deposition exhibited higher chromatin accessibility in BHB-treated spikelets compared to the CK (Supplementary Fig. 13c), suggesting that H3K9bhb regulates transcription by influencing chromatin accessibility. Furthermore, 117 genes showed increased H3K9bhb deposition, transcription, and chromatin accessibility, including genes involved in immunity (Supplementary Fig. 13d and Supplementary Data 5). In addition, we generated a CRISPR/Cas9 mutant of *WALL-ASSOCIATED RECEPTOR-LIKE KINASE 90* (*OsWAK90*), which was speculated to be a positive regulator of plant pathogen resistance in a previous study[29] and exhibited both

hyper-H3K9bhb levels and upregulated transcription here (Supplementary Fig. 12). The *wak90* mutant exhibited higher susceptibility to rice false smut (Fig. 5k) and rice blast (Fig. 5l), further suggesting the involvement of BHB-stimulated genes in plant defense mechanisms.

## Dynamic changes of H3K9bhb in rice in response to fungal infection

To further explore the involvement of BHB in regulating host immunity during fungal infection, BHB concentrations in infected and uninfected rice spikelets at 1 day post inoculation (dpi) were analyzed using LC–MS/MS. Production of BHB by the host during fungal infection was significantly increased (Fig. 6a). To investigate the functional roles of H3K9bhb and H3K9ac in plant responses to rice pathogens, we examined the levels of these marks on histones isolated from rice spikelets infected with *U. virens* and uninfected rice spikelets (CK) of the susceptible cultivar 'Wanxian-98' at 1 dpi. H3K9bhb level was substantially higher upon *U. virens* infection, while H3K9ac level remained unchanged (Fig. 6b). This finding suggests that H3K9bhb may have a key role in plant defense mechanisms. To investigate the function of H3K9bhb in regulating the transcription of defense-related genes during fungus infection, we conducted H3K9bhb ChIP-seq and RNA-seq (Supplementary Fig. 4) on infected rice spikelets and uninfected rice spikelets at 1 dpi. The ChIP-seq analysis revealed a significant increase in overall H3K9bhb levels in infected samples compared with CK samples (Fig. 6c, d), consistent with the western-blot results (Fig. 6b). A differential analysis of the ChIP-seq data showed that 3420 genes were associated with an increased deposition of H3K9bhb, while 478 genes showed a decreased deposition of this mark (>1.5 fold, *FDR* < 0.05; Supplementary Data 6) after *U. virens* infection (Fig. 6e). The upregulated genes were mainly enriched in the response to stimulus, phosphorylation, and the response to stress, further suggesting that H3K9bhb is involved in the plant stress response (Fig. 6f). In parallel, an RNA-seq analysis identified 2135 upregulated genes (>2-fold, *P* < 0.05) and 1296 downregulated genes (>2-fold, *P* < 0.05; Supplementary Data 7) in infected rice spikelets compared with CK rice spikelets (Fig. 6g). A GO pathway analysis showed that the upregulated genes were mainly enriched in the responses to chitin, fungi, and jasmonic acid pathways (Fig. 6h). Correlation analysis revealed a moderate positive correlation ($r = 0.3$, values between 0.3 and 0.7 indicate a moderate correlation) between the changes in gene expression and H3K9bhb levels (Fig. 6i). Notably, genes that were upregulated in infected rice spikelets also exhibited an increase in H3K9bhb compared with CK rice spikelets (Fig. 6j). A Venn diagram demonstrated that 238 genes exhibited both upregulation and hyper-H3K9bhb modifications in response to infection (Fig. 6k and Supplementary Data 8), suggesting that their expression was upregulated by the histone mark. Subsequent RT-qPCR and ChIP-qPCR assays performed on four selected genes confirmed their upregulated gene transcription and increased H3K9bhb levels in infected rice spikelets (Supplementary Figs. 14 and 15). The 238 genes that were upregulated and had hyper-H3K9bhb levels following infection were mainly enriched in the responses to stimulus and stress, as well as the defense-response pathways (Fig. 6l). Further analysis of the expression of these genes in rice under different pathogen treatments revealed that most were upregulated by the various pathogen treatments (Supplementary

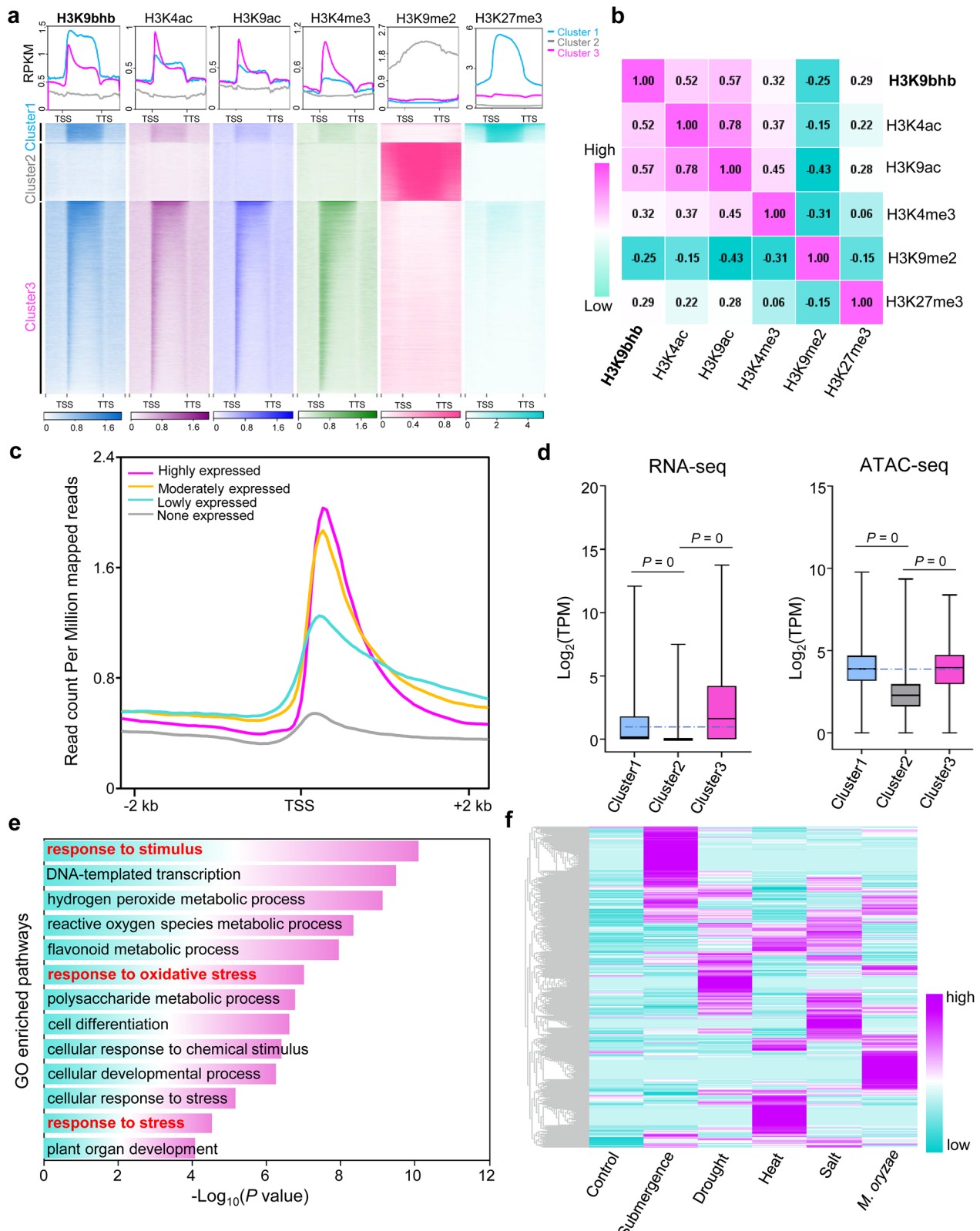

**Fig. 3 | Comparative analysis of H3K9bhb and other histone modifications in rice. a** Metaplots and heatmaps of H3K9bhb and other chromatin modification profiles in the wild-type background. RPKM: reads per kilobase per million mapped reads. **b** Correlation between H3K9bhb and other chromatin modifications. **c** H3K9bhb signal intensity in genes grouped into four categories based on their RNA-seq counts: top, middle, low, and bottom quartiles. TSS: transcriptional start site. **d** Left: Box plots showing gene expression levels of clustered genes from (**a**).

Right: Box plots showing chromatin opening state levels of clustered genes from (**a**). TPM: transcripts per kilobase million. In the box plots, horizontal lines represent medians, and box limits indicate the 25th and 75th percentiles. *P-values* were determined by a two-tailed, paired Student's *t*test. **e** GO pathway enrichment analysis of cluster I genes from (**a**). **f** Expression analysis of cluster I genes from (**a**) under different stress treatments.

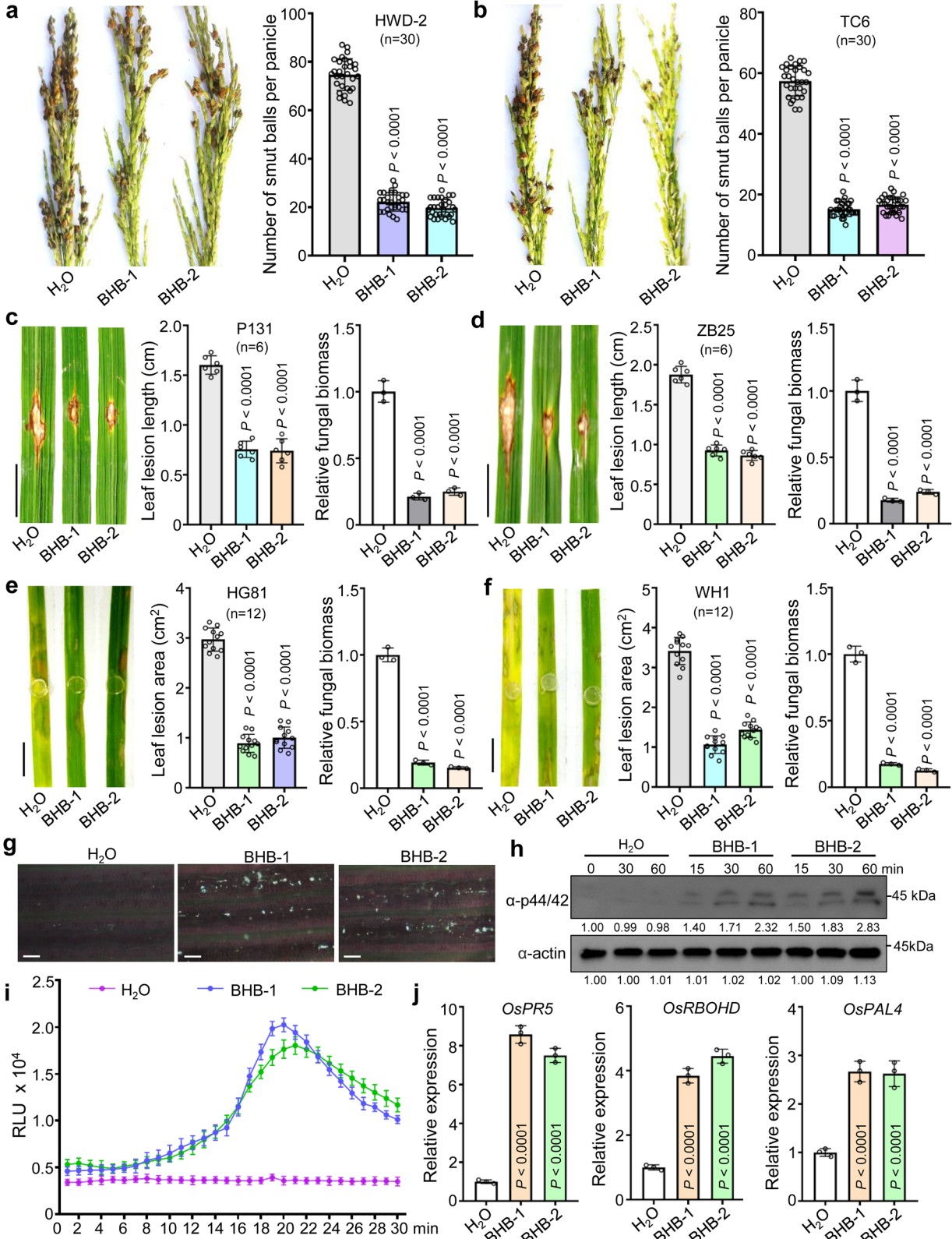

Fig. 16). Collectively, these findings support the idea that H3K9bhb has a role in regulating the disease response in rice.

## Identification and functional characterization of HDACs in the regulation of H3K9bhb

In mammals, some $Zn^{2+}$-dependent HDACs and $NAD^+$-dependent sirtuins have deacylase activity; for example, SRT7 and SRT5 exhibit desuccinylase activity[30,31], while HDAC1–3 display robust delactylase and decrotonylase activities[32,33]. Given the deacylase activities of HDACs, we analyzed the HDAC mutants in rice to determine whether histone acetylases were involved in the removal of histone Kbhb. Immunoblotting with anti-Kbhb and anti-H3K9bhb antibodies revealed that, among various rice HDAC mutants tested, srt1, srt2, and hda705 exhibited the most significant increases in global histone Kbhb

**Fig. 4 | Resistance assay of water (H₂O) and 100 µM BHB-treated rice plants against rice pathogens. a**, **b** Resistance assay of H₂O and BHB-treated rice spikelets against *U. virens* strains HWD-2 and TC6 at 21 dpi, data were collected from three independent experiments for each treatment with ten panicles. **c**, **d** Disease symptoms and leaf lesions of H₂O and BHB-treated rice leaves at 14 dpi following spot inoculation with *M. oryzae* strains P131 and ZB25. The leaf lesion length was measured using ImageJ software, data are means ± SD from six leaves. Relative fungal biomass was determined using RT-qPCR for *M. oryzae MoPot2* and normalized to rice *OsUBQ1*, data are means ± SD from three replicates. **e**, **f** Disease symptoms and leaf lesions of H₂O and BHB-treated rice leaves at 3 dpi with *R. solani* strain HG81 or WH1. The leaf lesion area was measured using ImageJ software. Data were collected from three independent experiments for each treatment, with four leaves. Relative fungal biomass was determined using RT-qPCR for *R. solani RsPal1* and normalized to rice *OsUBQ1*, data are means ± SD from three replicates. **g** Callose deposition assay in rice leaves. Rice leaves were treated with H₂O and BHB

for 24 h, followed by staining with aniline blue. Scar bars, 100 µm. **h** Activation of MAPK signaling in rice leaves (treated with H₂O or BHB). MAPK activation was detected by immunoblotting with the phospho-p44/42 MAPK antibody at the indicated times. Actin was used as the loading control. The immunoblot signals were quantified using ImageJ. Relative signals (with the phosphorylation levels of MPK3 and MPK6 at 0 min set as 1.00) are indicated below the bands. Images shown are representative of two independent experiments. **i** ROS burst in rice leaves treated with H₂O or BHB. RLU, relative luminescence unit. Data are means ± SD from three independent experiments. **j** Relative transcript levels of *OsPR5*, *OsRBOHD*, and *OsPAL4* in rice plants treated with H₂O and BHB detected by RT-qPCR, data are means ± SD from three replicates. For figures (**c**–**f**), the bar represents 1 cm. Data in (**a**–**f**, **j**) are analyzed by one-way ANOVA followed by two-sided LSD test for multiple comparisons, and the adjusted *P*-values were shown. Source data are provided as a Source Data file.

levels (Fig. 7a), suggesting their potential roles in histone Kbhb removal. Based on these immunoblot results, we selected these three HDACs for further characterization. An in vitro assay using the three recombinant HDACs (SRT1, SRT2, and HDA705) and rice-derived histones demonstrated that only SRT2 and HDA705 had notable de-Kbhb activity (Fig. 7b). The de-Kbhb activity of HDA705 was further validated by H3K9bhb assays in *HDA705* overexpression plants and by in vitro enzyme activity assays of both HDA705 and HDA705^H154A (Supplementary Fig. 17). HDA705 plays important roles in plant disease resistance[7] (Supplementary Fig. 18), and we sought to determine whether this function is mediated through H3K9bhb. To investigate this, we performed H3K9bhb ChIP-seq in *hda705* mutant and wild-type (WT) plants (Supplementary Fig. 19a). ChIP-seq analysis revealed that the loss of *HDA705* led to a significant genome-wide increase in H3K9bhb deposition (Fig. 7c, d), confirming the histone de-Kbhb activity of HDA705. A total of 30,267 H3K9bhb-marked genes were identified in WT plants, while 31,703 were detected in *hda705* mutants (Supplementary Fig. 20). Among them, 28,746 genes were commonly marked in both genotypes (Supplementary Fig. 20). Compared with WT, *hda705* exhibited a gain of 2957 and a loss of 1521 H3K9bhb-marked genes (Supplementary Fig. 20). Further statistical analysis showed that 2971 genes displayed significantly increased H3K9bhb levels in *hda705*, whereas 1048 genes exhibited significantly decreased H3K9bhb levels (>1.5-fold change, *FDR* < 0.05; Fig. 7e and Supplementary Data 9). A GO analysis indicated that the genes upregulated in *hda705* were enriched in various pathways, including the defense response, responses to bacteria and salicylic acid (Fig. 7f). ChIP-qPCR and RT-qPCR analyses in *hda705* (Fig. 7g) and overexpression plants (Supplementary Fig. 21) confirmed the direct regulation of HDA705 on H3K9bhb and the transcription of genes involved in disease resistance[34–37]. In addition, HDA705 was found to have H3K9ac deacetylation activity[38], prompting us to test the H3K9ac acetylation levels of these genes, which showed no significant difference (Supplementary Fig. 22). Next, we also performed H3K9ac ChIP-seq in *hda705* and WT plants (Supplementary Fig. 19b). Comparative data analysis showed that 2258 genes had significantly increased H3K9ac levels in *hda705* compared to the WT, while 854 genes exhibited significantly decreased H3K9ac levels (>1.5-fold, *FDR* < 0.05; Supplementary Fig. 23a and Supplementary Data 10). GO pathway analysis indicated that genes with increased H3K9ac deposition in *hda705* were enriched in various pathways, such as diterpenoid metabolic process, gibberellin metabolic process, response to sucrose, regulation of hormone levels, and seedling development (Supplementary Fig. 23b), suggesting that HDA705-regulated H3K9ac genes are primarily involved in metabolic processes and development and are functionally distinct from H3K9bhb-regulated genes. This was further confirmed by the observation that few overlapping genes were found between H3K9ac hyper-marked genes and H3K9bhb hyper-marked genes (Supplementary Fig. 23c). In addition, we observed a moderate negative

correlation (*r* = −0.3, values between 0.3 and 0.7 indicate a moderate correlation) between H3K9ac and H3K9bhb changes in the genes (*N* = 13,881) that exhibited opposite dynamic changes (i.e., increased H3K9ac with decreased H3K9bhb, or vice versa) (Supplementary Fig. 23d), confirming the previous notion that H3K9ac and H3K9bhb co-occupy the same genomic loci in a mutually exclusive manner. Given that the *srt2* mutant also exhibited increased resistance to plant pathogens[9] (Supplementary Fig. 18b), we next investigated whether *SRT2*-regulated H3K9bhb contributes to plant immunity. To test this, we performed H3K9bhb ChIP-seq in both *srt2* mutant and WT plants to explore its role in histone Kbhb regulation (Supplementary Fig. 19c). ChIP-seq data analysis showed that *srt2* mutants exhibited increased genome-wide H3K9bhb levels (Supplementary Fig. 24a), verifying its histone de-Kbhb activity. A total of 30,863 H3K9bhb-marked genes were identified in WT plants, while 31,210 were detected in *srt2* mutants (Supplementary Fig. 24b). Among them, 27,982 genes were commonly marked in both genotypes (Supplementary Fig. 24b). Compared with WT, *srt2* exhibited a gain of 3228 and a loss of 2881 H3K9bhb-marked genes (Supplementary Fig. 24b). Further statistical analysis identified 2724 genes with significantly increased H3K9bhb levels and 2002 genes with significantly decreased H3K9bhb levels in *srt2* compared to WT (>1.5-fold change, *FDR* < 0.05; Supplementary Fig. 24c; Supplementary Data 11). GO analysis revealed that genes with increased H3K9bhb levels in *srt2* were enriched in pathways related to photosynthesis, ribosome biogenesis, and response to water, among others (Supplementary Fig. 24d). However, no enrichment was observed in pathways associated with plant immunity, suggesting that the disease resistance of *srt2* might primarily be regulated by histone acetylation, as reported previously[9]. Collectively, these findings suggest that HDA705-mediated regulation of H3K9bhb plays a key role in the plant–pathogen response.

**Proteome-wide identification of Kbhb substrates in rice flowers**
In animal cells, Kbhb have been detected in non-histone proteins[23]. To identify rice proteins with Kbhb modifications, we conducted a proteomics screening using mass spectrometry (Supplementary Fig. 25a). Three biological replicates were performed (Supplementary Fig. 25b). The peptide lengths peaked at 9–14 amino acids, and the average mass error was <5 ppm, suggesting that our mass data were of good quality (Supplementary Fig. 26). We detected 2159 Kbhb sites on 1128 proteins (Supplementary Fig. 25b and Supplementary Data 12). After excluding the sites identified only once, 2016 sites on 1070 proteins were identified (Supplementary Fig. 25b and Supplementary Data 12). These sites were considered reliable Kbhb sites (proteins) and were analyzed further.

The rice proteins exhibiting Kbhb had varying numbers of modification sites, ranging from one to five or more sites per protein (Supplementary Fig. 25c). Of the 1070 proteins identified, more than 50% only had one site, while 3.6% of proteins were heavily Kbhb-

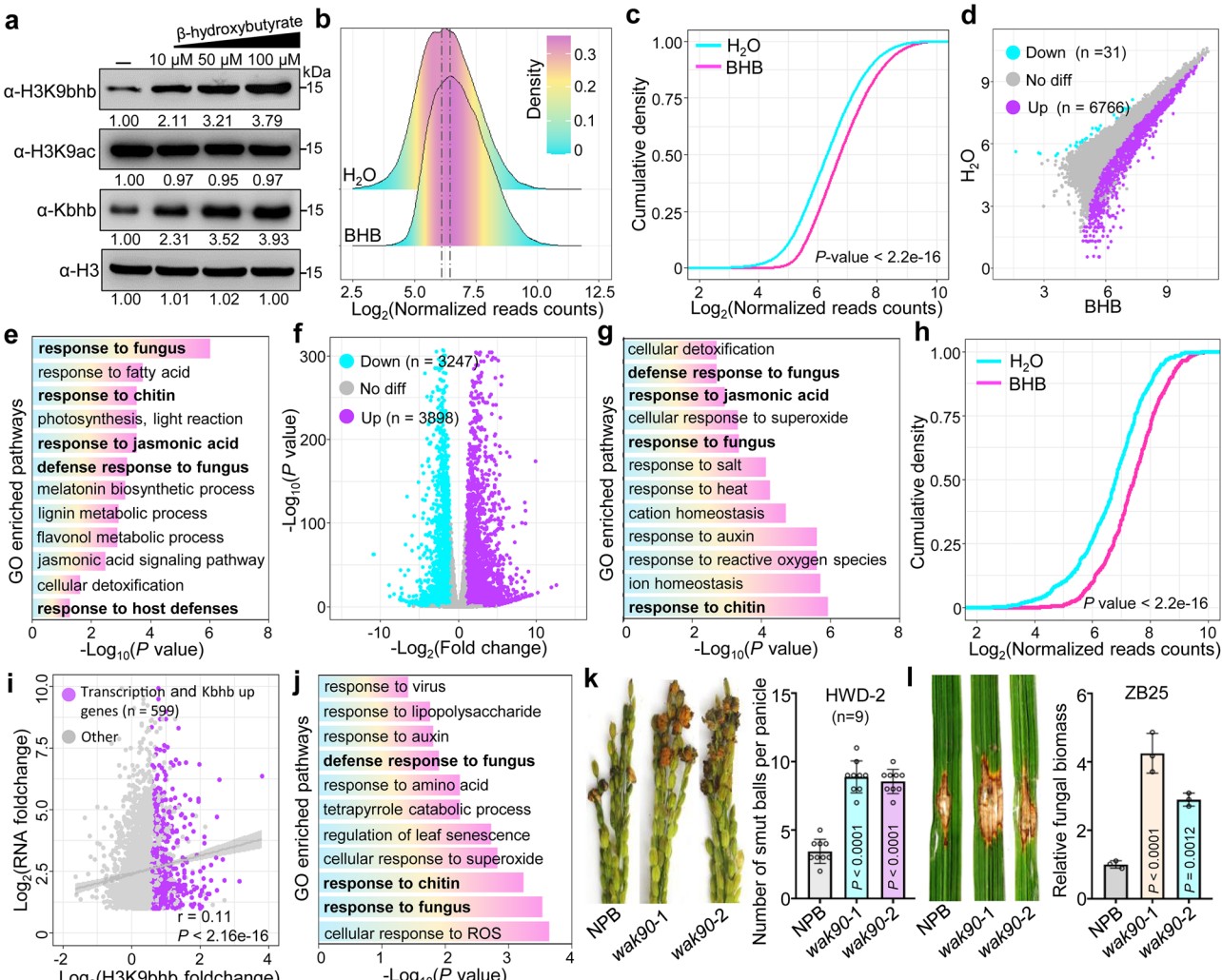

**Fig. 5 | Genome-wide analysis of H3K9bhb in rice spikelets treated with water (H2O) and 100 μM BHB. a** Analysis of histone Kbhb levels in rice spikelets treated with different concentrations of BHB. Histones were acid-extracted from the rice spikelets and immunoblotted with the indicated antibodies. The immunoblot signals were quantified using ImageJ. Relative quantified signals for each band are indicated, with the control loading set as 1.00. Images shown are representative of two independent experiments. **b** Ridge plots of H3K9bhb levels in rice spikelets treated with H2O and BHB. **c** Cumulative density plots of H3K9bhb levels in rice spikelets treated with H2O and BHB. **d** Scatter plot of ChIP-seq data showing the differential occupancy of H3K9bhb in rice spikelets treated with BHB relative to those treated with H2O. **e** GO enriched pathways of H3K9bhb significantly upregulated genes. **f** Volcano plots of differential transcript levels in rice spikelets treated with BHB and H2O. **g** GO pathway analysis of the genes (n = 3898) that were upregulated in BHB compared to H2O-treated rice spikelets. **h** Cumulative density plots of H3K9bhb levels for genes significantly upregulated in (**f**) in rice spikelets treated with H2O and BHB. **i** Correlation analysis of expression changes and H3K9bhb changes in BHB-treated versus H2O-treated rice spikelets. The person correlation coefficient is shown. *P-values* were determined by a two-tailed, paired Student's *t*test. **j** GO-enriched pathways of genes (n = 599) that exhibit hyper-H3K9bhb and significantly upregulated expression levels. **k** Resistance assay of the wide-type NPB and *wak90* mutant plants against *U. virens* HWD-2 at 21 dpi, data are means ± SD from nine panicles. **l** Disease symptoms and leaf lesions of the wild-type NPB and *wak90* mutant plants at 14 dpi following spot inoculation with *M. oryzae* ZB25. Relative fungal biomass was determined using RT-qPCR for *M. oryzae MoPot2* and normalized to rice *OsUBQ1*, data are means ± SD from three replicates. The *P*-values in (**c**, **h**) were calculated using a two-sample Kolmogorov–Smirnov test. The read counts (in **b**, **c**, and **h**) were normalized using the DESeq2 size factor normalization method. Data in (**k**, **l**) are analyzed by one-way ANOVA followed by two-sided LSD test for multiple comparisons, and the adjusted *P*-values were shown. Source data are provided as a Source Data file.

modified (>5 sites), including the Rubisco large subunit, HSP81-2, peroxidase, and glyceraldehyde-3-phosphate dehydrogenase (GAPDH) (Supplementary Fig. 27). We also detected 26 histone Kbhb sites, including several Kbhb sites on histones H3 and H4, as well as multiple Kbhb sites on histones H2A and H2B (Supplementary Data 13). In addition, 20 transcription factors and five chromatin regulators were identified as Kbhb-modified (Supplementary Data 13). Subcellular localization analysis revealed that Kbhb-modified proteins were primarily distributed in the chloroplasts (40.5%), cytoplasm (31.6%), and nucleus (15.0%) (Supplementary Fig. 25d). We identified conserved sequence motifs in different subdomains, with leucine (L),

phenylalanine (F), and tyrosine (Y) most commonly found at the +1 position flanking the Kbhb site and glutamate (E) at the −1 position (Supplementary Fig. 28). To identify the pathways regulated by the Kbhb proteins, we performed a GO enrichment and Kyoto Encyclopedia of Genes and Genomes (KEGG) pathway analyses. The Kbhb protein functions were mostly enriched in biological processes such as metabolic and cellular processes, molecular functions such as binding and catalytic activities, and cellular components such as protein-containing complexes and cellular anatomical entities (Supplementary Fig. 25e). This finding suggests that Kbhb proteins play important roles in plant metabolism. The KEGG pathway analysis revealed that

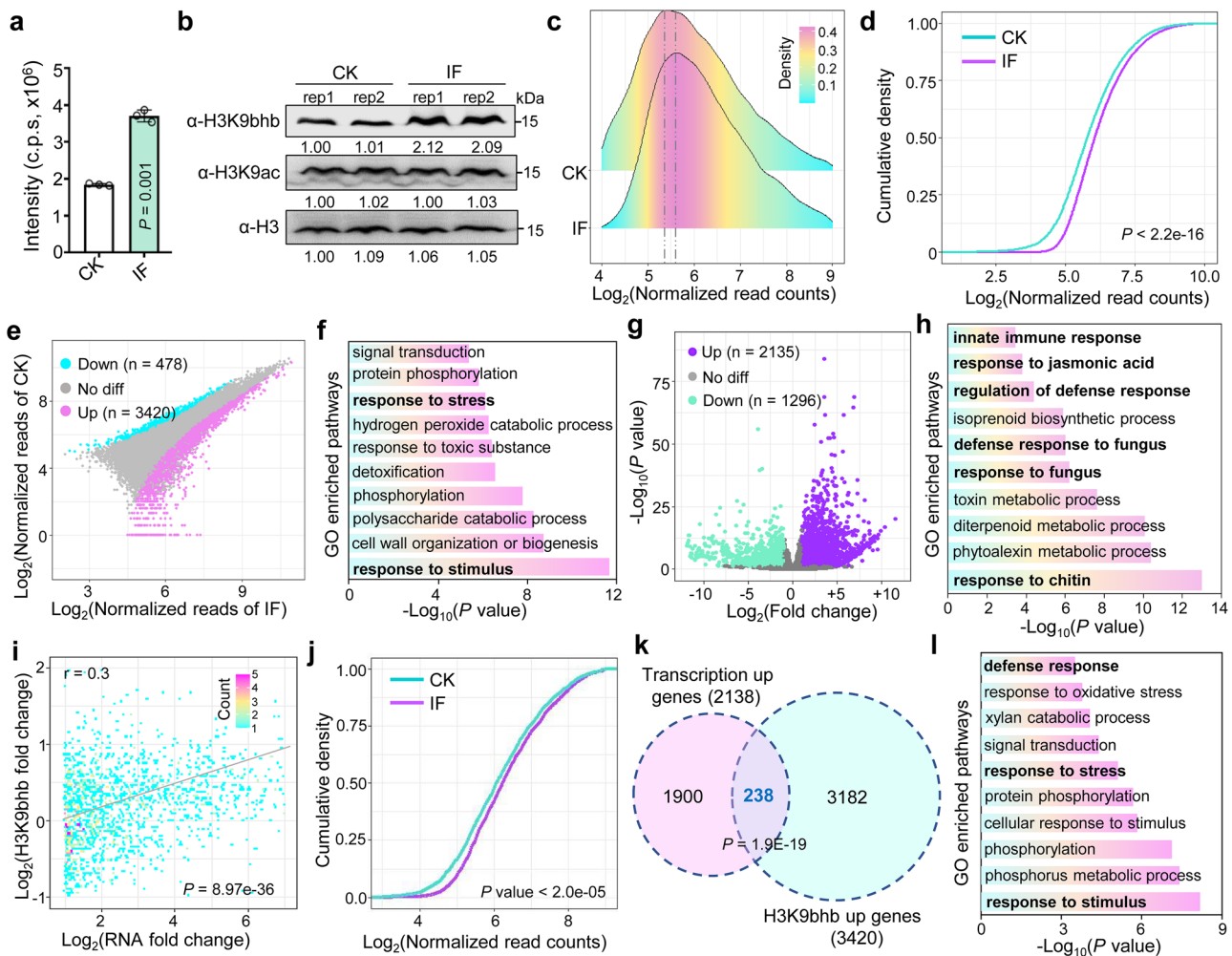

**Fig. 6 | Genome-wide analysis of H3K9bhb in *U. virens*-infected (IF) and uninfected (CK) rice spikelets at 1 dpi. a** β-Hydroxybutyrate concentration assay in IF and CK rice spikelets using liquid chromatography-tandem mass spectrometry. Data from three independent experiments are shown as mean ± SD. *P*-value was calculated by a two-tailed, paired Student *t* test. **b** Analysis of H3K9bhb and H3K9ac levels in IF rice spikelets compared with CK rice spikelets by immunoblotting. The histone modification antibodies used in the tests are indicated on the left. The immunoblot signals were quantified using ImageJ. Relative signals (with the CK rep1 signal set at 1.00) are indicated below the bands. Images shown are representative of two independent experiments. **c** Ridge plots of H3K9bhb levels in CK and IF rice spikelets. **d** Cumulative density plots of H3K9bhb levels in IF and CK rice spikelets. **e** Scatter plot of ChIP-seq data showing the differential occupancy of H3K9bhb in IF relative to CK rice spikelets. Peaks with fold changes >1.5 and *FDR* values <0.05 were considered upregulated or downregulated. **f** GO enriched pathways of H3K9bhb significantly upregulated genes. **g** Volcano plots of differential transcript levels in IF relative to CK rice spikelets. **h** GO pathway analysis of the genes (*n* = 2135) that were upregulated in IF rice spikelets in comparison to CK. **i** Correlation analysis of expression changes and H3K9bhb changes in IF versus CK rice spikelets. Person correlation coefficient is shown. *P-values* were determined by a two-tailed, paired Student's *t* test. **j** Cumulative density plots of H3K9bhb levels for genes significantly upregulated in (**g**) in CK and IF rice spikelets. **k** Venn diagrams showing the overlap between hyper-H3K9bhb genes and transcriptionally upregulated genes in IF versus CK rice spikelets. *P*-value was calculated by Fisher's exact test. **l** GO-enriched pathways of genes (*n* = 238) that are both H3K9bhb and transcriptionally upregulated in IF versus CK rice spikelets. The *P*-value in (**d**, **j**) were calculated using a two-sample Kolmogorov–Smirnov test. The read counts in (**c**, **d**, and **j**) were normalized using the DESeq2 size factor normalization method. Source data are provided as a Source Data file.

ribosome, glyoxylate and dicarboxylate metabolism, glycolysis/gluconeogenesis, photosynthesis, pyruvate metabolism, and tricarboxylic acid cycles were the most enriched pathways (Supplementary Fig. 25f), which was consistent with the protein–protein interaction network (Supplementary Fig. 29). Collectively, these findings suggest that Kbhb is a complex PTM involved in diverse cellular functions.

## Discussion

Among the various histone Kbhb modifications in animal cells, H3K9bhb is the most extensively characterized[39–41]. Genome-wide profiling has shown that H3K9bhb is an active positive histone mark associated with genes involved in stress-response pathways, such as the starvation response, immune response, and depression in animal cells[10,24,25]. Despite its well-characterized functions in animal cells, the

functional roles of H3K9bhb in plants were unknown. In the present study, our data implies that H3K9bhb modification predominantly occurred in genic regions, with the highest enrichment observed at the TSSs of non-TE genes (Fig. 1). This pattern is consistent with observations in animal cells[10], suggesting a conserved role for H3K9bhb in the regulation of gene expression. The distribution of H3K9bhb closely correlates with other active histone marks, such as H3K4ac, H3K9ac, and H3K4me3 (Fig. 3), suggesting that H3K9bhb functions as an active histone modification. Previous studies have indicated a potential link between histone Kbhb and chromatin accessibility[42,43]. Therefore, it is hypothesized that Kbhb, similar to acetylation, may neutralize the positive charge on histones, thereby weakening histone-DNA interactions and leading to a more open chromatin state that promotes gene expression. However, the direct relationship between histone Kbhb

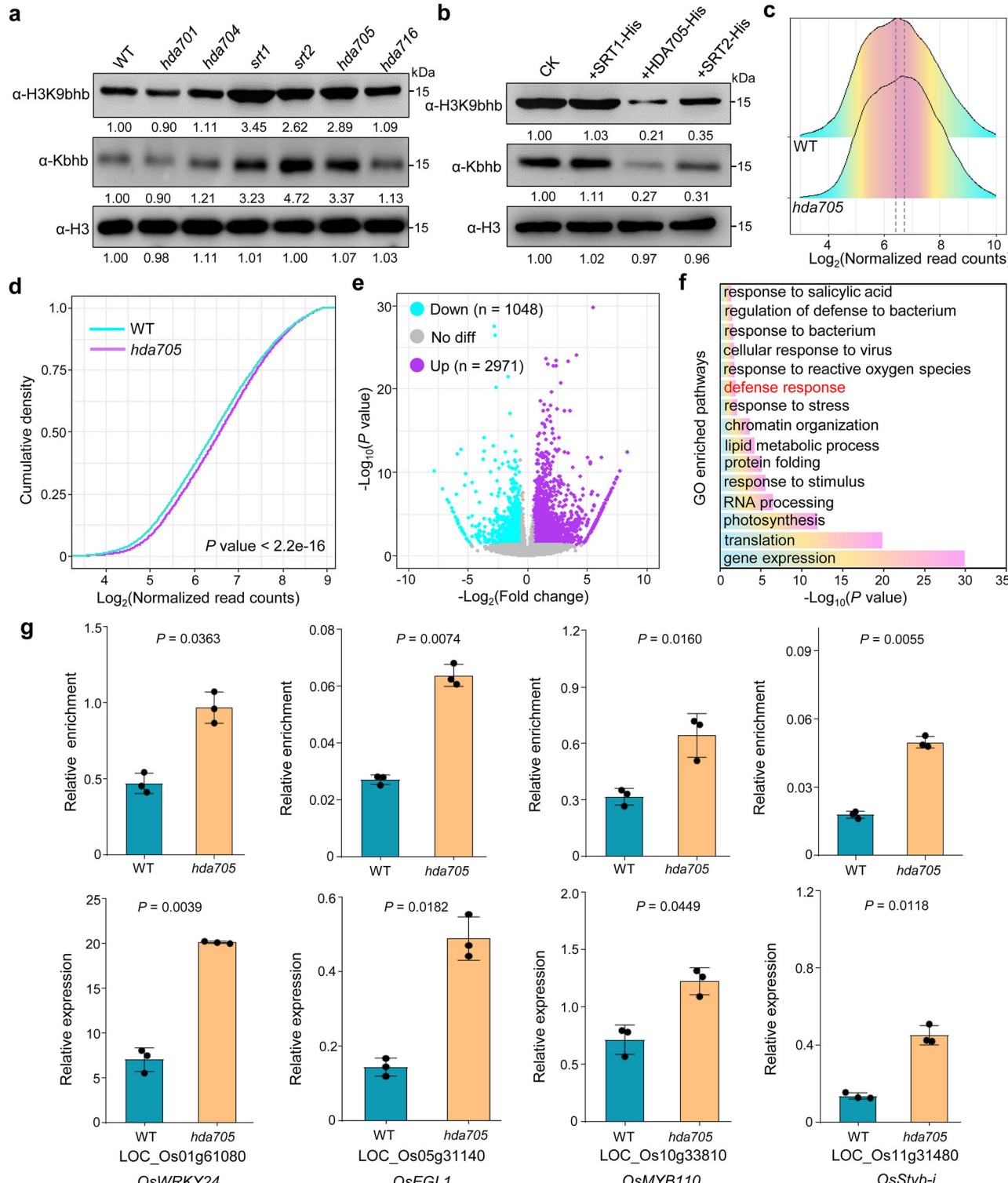

**Fig. 7 | SRT1, SRT2, and HDA705 have histone de-β-hydroxybutyrylase activities. a** Histone H3K9bhb levels in *hda701*, *hda704*, *hda705*, *hda716*, *srt1*, and *srt2* CRISPR/Cas9 plants compared to wild-type (WT) plants. **b** In vitro de-β-hydroxybutyrylatase activity assay. Histone H3 was used as a loading control. For figures (**a**, **b**), histones were acid-extracted from the rice leaves and immunoblotted with the indicated antibodies. Images shown are representative of two independent experiments. The immunoblot signals were quantified using ImageJ. Relative quantified signals for each band are indicated, with the first WT/CK loading set as 1.00. **c** Ridge plots of H3K9bhb levels in *hda705* and WT plants. The read counts were normalized using the DESeq2 size factor normalization method. **d** Cumulative density plots of H3K9bhb levels in *hda705* and WT plants. *P*-value was calculated

from a two-sample Kolmogorov–Smirnov test. The read counts were normalized using the DESeq2 size factor normalization method. **e** Volcano plots of differential H3K9bhb levels in *hda705* relative to WT. **f** GO pathway analysis of the genes (*n* = 2971) with significantly upregulated H3K9bhb levels in *hda705* in comparison to WT. **g** Assay of expression and H3K9bhb levels of genes in WT and *hda705* rice plants. Upper: H3K9bhb ChIP-qPCR assays of chromatin isolated from CK and *hda705* rice plants. Lower: RT-qPCR analysis of transcript levels in four selected genes in WT and *hda705* plants. Bars indicate means ± SD from three replicates. *P*-value was calculated by a two-tailed, paired Student *t* test. Source data are provided as a Source Data file.

and chromatin accessibility has not been extensively investigated. To address this, we performed ATAC-seq analysis, which revealed that hyper-Kbhb levels are associated with increased chromatin accessibility (Supplementary Fig 13), supporting its role in chromatin remodeling. In addition, we identified a subset of genes that were highly enriched in both H3K9bhb and H3K27me3 (Fig. 3a), a repressive mark, indicating a potentially bivalent chromatin modification state. In plants, bivalent modifications are involved in the activation of stress-response genes[44]. The relatively high chromatin accessibility and low gene expression levels suggest a poised state for the transcriptional activation of the co-marked genes, which was further supported by a gene expression analysis under various stress conditions (Fig. 3f).

Both H3K9bhb and H3K9ac occupy the same histone loci, suggesting a competitive relationship between these modifications. This competition was confirmed by ChIP-reChIP-qPCR analysis (Supplementary Fig. 7), the distinct functional roles of the two marks (Fig. 2d), and a negative correlation ($r = 0.3$) between their dynamic changes (Supplementary Fig. 23d), resembling the previously reported antagonism between histone H4K5/K8 acetylation and butyrylation in animal cells[45]. The considerable overlap of genes marked by H3K9bhb and H3K9ac may be explained by the rapid turnover of acetylation and β-hydroxybutyrylation on H3K9; however, it cannot be ruled out that H3K9bhb and H3K9ac may replace each other in different rice cells, as this study utilized mixed cells from the same tissue. Future studies employing single-cell epigenomic profiling may provide further insights into this phenomenon. In addition, we observed non-redundant functional roles for H3K9bhb and H3K9ac. Specifically, genes marked solely by H3K9bhb were predominantly associated with stress-response pathways, especially in the response to biotic stresses, while genes marked by H3K9ac alone were mainly enriched in metabolic and developmental processes (Fig. 2). Moreover, H3K9bhb was specifically induced under *U. virens* treatment, and HDA705-regulated H3K9ac was mainly enriched in metabolic processes and development, whereas HDA705-regulated H3K9bhb was associated with pathways related to plant immunity (Figs. 6, 7; Supplementary Figs. 14, 15, 22, 23). These suggest that H3K9bhb may play a distinct role under stress conditions where histone acetylation may not be as effective. The stress-specific response of H3K9bhb compared to H3K9ac was consistent with observations in animal cells[10,24,25], suggesting the conserved function of this mark in stress responses.

Emerging evidence suggests that histone marks play important roles in the metabolic control of epigenetics[46]. In animal cells, BHB acts as a signaling metabolite with various direct effects, including competitively inhibiting HDAC enzymes, binding to cell surface receptors, modulating ion channel activity, and post-translationally modifying proteins[47]. In humans, the appropriate supplementation of BHB has beneficial and life-preserving effects, including in the treatment of inflammation, cancer, epilepsy, dementia, and other neurological conditions[48–50]. In plants, we found that BHB enhanced rice resistance against three major fungal diseases (rice blast, rice false smut, and rice sheath blight) and could induce plant immunity (Fig. 4). Mechanistically, BHB may epigenetically regulate the chromatin accessibility of defense-related genes by inducing the H3K9bhb modification of their associated histones, leading to increased gene expression (Fig. 5 and Supplementary Fig. 13). This enhanced gene expression of defense genes may be regulated by plant immunity-related transcription factors, such as WRKY or NAC, as their binding motifs were found in the promoters of H3K9bhb-marked genes (Supplementary Fig. 5). BHB can be metabolized into β-hydroxybutyryl-CoA, and future work will explore the molecular regulation of its biosynthesis. Developing high-BHB plants with the associated improved resistance against multiple pathogens could be a promising biotechnological strategy for future crop breeding.

Several defense-related proteins play important roles in rice systemic immunity[51]. In this study, we revealed a substantial increase in H3K9bhb levels upon *U. virens* infection (Fig. 6), particularly at defense-related genes *OsWRKY70*, *1-AMINOCYCLOPROPANE-1-CARBOXYLIC ACID SYNTHASE 2 (OsACS2)*, *PHENYLALANINE AMMONIA-LYASE 4 (OsPAL4)*, and *ETHYLENE RESPONSE FACTOR 83 (OsERF83)*[52–56], which was associated with increases in their expression (Supplementary Fig. 14). This result indicates that the increase in H3K9bhb due to *U. virens* infection may enhance the expression of several defense-related genes, thereby boosting plant resistance.

Functional characterization of histone Kbhb marks requires the identification of the regulatory enzymes responsible for adding and removing this modification. Classically annotated HDACs were previously identified based on their ability to carry out acetylation and other acetylation-independent acylation reactions[30,43,57,58]. Studies conducted in animal cells have demonstrated that RPD3 and Sir2 family members possess de-Kbhb activities, while CBP/p300 acts as a Kbhb transferase[23]. Here, we found that HDA705, SRT1, and SRT2 functioned as erasers of Kbhb and exhibited corresponding enzymatic activity in vivo; however, only SRT2 and HDA705 were able to directly remove histone Kbhb in an in vitro activity assay (Fig. 7). The lack of significant de-Kbhb activity for SRT1 may be due to a requirement for an activating partner within a multi-protein complex, as has been shown for the in vivo activity of certain HDACs[59,60]. In addition, we observed that H3K9bhb may also be involved in rice resistance to pathogens in the *hda705* mutant. As HDA705 is also involved in the regulation of histone 2-hydroxyisobutyrylation[7,61], further investigation is needed to determine the substrate preference of HDA705 and elucidate the functional differences between these two histone acylations in plant immune responses.

Lysine acylations are also widespread in non-histone proteins located within various organelles[23]. In this study, we identified 2159 Kbhb sites on 1128 proteins across the rice proteome, which represents a global view of the Kbhb proteomic dataset in plants. Kbhb substrates were particularly enriched in functions related to ribosomes, glyoxylate and dicarboxylate metabolism, glycolysis/gluconeogenesis, and photosynthesis (Supplementary Fig. 25). This enrichment differs from that of Kbhb proteins in animal cells, which are primarily involved in pathways within the nucleus, such as RNA metabolism, chromatin organization, and DNA repair[23]. These findings suggest that there may be functional differences in Kbhb between animals and plants. We observed that a number of proteins involved in immune responses and the responses to stimuli were modified by Kbhb, including MITOGEN-ACTIVATED PROTEIN KINASE 6 (MPK6) and MKK1, RESPIRATORY BURST OXIDASE (OsRBOHD), CATALASE (OsCatA and OsCatC), ATP-CITRATE SYNTHASE (OsACL2), and the disease-resistance protein SUPPRESSOR OF G2 ALLELE *skp1* (OsSGT1)[62–65]. The Kbhb-sensitive proteins identified here provide a valuable research resource for future mechanistic studies.

In conclusion, our study provides a comprehensive characterization of H3K9bhb in plants, highlighting its conserved role in the regulation of gene expression and its potential involvement in plant defense. Mechanistically, we found that BHB can induce plant immunity and epigenetically regulate the chromatin accessibility of H3K9bhb-stimulated defense-related genes. In addition, we identified erasers for histone Kbhb and presented the first Kbhb-protein dataset in plants. Collectively, these findings enhance our understanding of plant epigenetic regulation and lay the foundation for further exploration into the roles and mechanisms of histone β-hydroxybutyrylation.

## Methods
### Plant material and treatments
*Oryza sativa* cvs. 'WX98', 'Nipponbare' (NPB), and 'ZH11' plants were grown in a glasshouse. *HDA705*-overexpression (*HDA705-OE*) plants

were driven by the cauliflower mosaic virus 35S promoter, with a 3×FLAG tag fused to the N-terminus of the protein. The histone deacetylase mutants used in this study were previously characterized and reported in our earlier work[7]. Rice plants were inoculated with the fungal strains *Magnaporthe oryzae* ZB25 or P131, *Rhizoctonia solani* HG81 or WH1, and *U. virens* HWD-2 or TC6. Rice plants at the booting stage were inoculated with *U. virens* mycelial/spore suspensions ($1 \times 10^6$ conidia/mL) using a syringe, and the number of false smut balls was counted at 21 days post inoculation (dpi)[66]. For spot-inoculations, the leaves of 4-week-old rice plants were punctured with a needle, and a droplet of 10 μL *M. oryzae* conidial suspension ($1 \times 10^5$ conidia/mL) was placed at each wound site, the disease lesions were examined at 14 dpi[67], and the relative fungal biomass was determined using RT-qPCR for the *M. oryzae Pot2* gene normalized to rice *OsUBQ1*. The leaves of 4-week-old rice plants that were inoculated with *R. solani*, the lesion areas were scored at 3 dpi and calculated using ImageJ software (v1.6.0_24), and the relative fungal biomass was determined using RT-qPCR for the *R. solani Pal1* gene normalized to rice *OsUBQ1*. All inoculation experiments were repeated two times.

WX98 rice plants at the booting stage were inoculated with *U. virens* HWD-2 or 100 μM BHB for 1 dpi, and the rice spikelets were collected for ChIP-seq and RNA-seq analysis. The rice flowers were collected and used for the Kbhb proteome.

### Detection of ROS accumulation, MAPK activation, and callose deposition

Rice seedlings (cvs. 'WX98') were grown on Murashige and Skoog (MS) medium in the growth chamber for 12 d. Leaves of the seedlings were cut into disks (4 mm in diameter) and then submerged in distilled water in a 96-well plate in the light overnight. The distilled water was removed by pipetting and replaced with 100 μL per well of an assay solution (100 μM BHB, 10 μg/mL horseradish peroxidase, and 50 μM luminol) using a multi-channel pipette. Chemiluminescence was measured in 30 s intervals over a period of 30 min on a SPARK-10M microplate reader (TECAN, Switzerland). Ten biological replicates were used per sample. Distilled water was used as the mock control.

The leaves of 2-week-old rice seedlings (cvs. 'WX98') were treated with 100 μM BHB or water with vacuuming for 15, 30 and 60 min, total proteins were extracted and then analyzed by immunoblotting using the anti-Phospho-p44/42 (1:1000 dilution, 9101S, Cell Signaling Technology, USA) and anti-actin (1:2000 dilution, AC009, ABclonal, China). The intensity of the immunoblotting bands was analyzed using Image J software (v1.6.0_24).

The leaves of 2-week-old rice seedlings (cvs. 'WX98') were treated with 100 μM BHB with vacuuming for 30 min and then incubated at 28 °C for 24 h. The leaves were fixed in ethanol:acetic acid (3:1, v/v) solution for 6 h with frequent changes with fresh solution. After three washes with water, the leaves were incubated in staining solution (0.01% [w/v] aniline blue, 150 mM $K_2HPO_4$) for 4 h on an end-over-end shaker. The leaves were then observed using an LSCM under UV light (340– 380 nm).

### BHB measurement

For BHB metabolite measurements, samples were collected separately and freeze-dried. The freeze-dried tissue was dissolved in 50 mL of 80% methanol, sonicated for 30 min, and allowed to stand overnight. The mixture was then centrifuged at $13,400 \times g$ and 4 °C for 10 min, and the supernatant was collected. The sample was filtered using a microporous membrane, and the filtrate was stored in an LC-MS compatible injection vial. HPLC analysis was performed using a Waters ACQUITY UPLC HSS T3 C18 column (1.8 μm, 2.1 mm × 100 mm). Samples were eluted with a linear gradient starting at 90% mobile phase B (95% acetonitrile in water containing 5 mM ammonium acetate, pH 6.8) in mobile phase A (water containing 5 mM ammonium acetate, pH 6.8), at a flow rate of 900 nL/min over a 20 min runtime. Eluted BHB was

introduced via electrospray ionization into an LC-ESI-MS/MS system (Shim-pack UFLC Shimadzu CBM20A) operating in targeted ion mode. Labsolution Insight LCMS software was used for peak shape correction and integration. Both intelligent integration algorithms and manual correction methods were applied to obtain peak areas and calculate relative metabolite content.

### Histone extraction and immunoblot analysis

Histones from rice, Arabidopsis, maize, and tobacco plants were extracted using the commercial histone extraction kit (Epigentek USA, OP-0006-100) following the manufacturer's protocol. The extracted histones were first denatured at 95 °C for 10 min and then analyzed by immunoblotting using the following antibodies: anti-H3K9ac (1:1000 dilution, Millipore, 07-352), anti-H3 (1:1000 dilution, Abcam, ab1791), anti-Kbhb (1:1000 dilution, PTM, PTM-1201RM), and anti-H3K9bhb (1:1000 dilution, PTM, PTM-1250). The secondary antibody used was peroxidase-conjugated goat anti-rabbit antibody (1:10000 dilution, Abbkine, A21020). The intensity of the immunoblotting bands was analyzed using Image J software (v1.6.0_24).

### ChIP-seq and data analysis

The chromatin immunoprecipitation (ChIP) experiment was performed as previously reported. In brief, rice spikelets were first crosslinked in 1% formaldehyde (F8775, Sigma-Aldrich) with a protein inhibitor (SJ-MK0001, Shandong Sparkjade Biotechnology Co., Ltd.) for 30 min, and then ground into powder in liquid nitrogen. The chromatin was then extracted and sonicated to 200–500 bp. Next, the chromatin was immunoprecipitated using the indicated antibodies. The immunoprecipitated chromatin was washed with wash buffers (low salt buffer [150 mM NaCl, 1 mM EDTA, 50 mM HEPES-KOH, 0.1% sodium deoxycholate, 1% Triton X-100, 0.1% SDS], high salt buffer [same as low salt buffer except 350 mM NaCl was used], LiCl buffer [10 mM Tris-HCl pH 8.0, 250 mM LiCl, 0.5% NP-40, 0.1% sodium deoxycholate, 1 mM EDTA], and TE buffer [1 mM EDTA, 10 mM Tris-HCl, pH 8.0]), followed by de-crosslinking with 5 M NaCl at 65 °C overnight and purification with chloroform. The purified DNA fragment was used to construct DNA libraries following the Illumina TruSeqChIP Sample Prep Set A kit protocol and sequenced on the Illumina NovaSeq 6000 using the PE 150 method. For data analysis, raw data were first cleaned using FastP (v0.232) to remove low-quality reads and adapters. Cleaned reads were then mapped to the rice genome using Bowtie2 (version 2.3.5.1) with default settings. Samtools (v1.9) was used to remove duplicated reads, and histone modification peaks were called using MACS software (version 2.2.7.1) with default parameters (-f BAMPE -B -q 0.05). Wig files generated by deepTools (v2.5.3) (BPM normalization) were visualized in IGV (version 2.3.88). Differential histone modification peaks were analyzed using DiffBind (v3.5) with default parameters. Peaks annotation was performed using annotatePeaks.pl of Homer (v4.11). Gene ontology analysis was conducted using clusterProfiler (4.10.0). Ridge plots, scatter plots, and heatmaps were generated using Tbtools (v0.6)[68] and R (v3.5), respectively.

### RT-qPCR and ChIP-qPCR analysis

Total RNA from rice plants was isolated using TRIzol reagent (Invitrogen, 15596018). 1 μg of RNA was used to generate complementary DNA using the reverse transcription kit (Vazyme, R212-01). Real-time PCR was performed on an ABI 7500 real-time PCR system using the SYBR Premix ExTaq (TaKaRa, RR820A) reagent. Actin was employed as an internal control for gene expression normalization. Three biological replicates were performed for each sample. Relative expression was measured using $2^{-\Delta\Delta CT}$ method[69]. For ChIP-qPCR, a chromatin immunoprecipitation experiment was carried out as stated in the "ChIP-seq and data analysis section". The purified immunoprecipitated DNA fragments were used for ChIP-qPCR analysis. Three biological replicates were performed using samples collected

from three independent cultures. The primers were listed in Supplementary Data 14.

## ChIP-re-ChIP-qPCR assay

Two grams of 14-day-old rice tissue were cross-linked with 1% formaldehyde for chromatin extraction. The chromatin was sonicated to ~200 bp fragments, centrifuged ($13400 \times g$, 10 min, 4 °C), and the supernatant was collected. A 20 μL portion was set aside as the input sample, while the rest was incubated with antibody-coated magnetic beads (Invitrogen, 10001D) for immunoprecipitation. The chromatin was sequentially washed with low-salt, high-salt, LiCl, and TE buffers, then eluted using ChIP elution buffer at 65 °C. The eluted chromatin underwent a second immunoprecipitation with antibody-coated beads, followed by the same wash and elution steps. The recovered protein-DNA complexes were de-crosslinked, and DNA was analyzed by RT-qPCR. Sequential ChIP used the following antibodies: anti-H3K9ac (1:1000 dilution, Millipore, 07-352), anti-anti-H3K4ac (1:1000 dilution, Abcam, ab176799), and anti-H3K9bhb (1:1000 dilution, PTM Bio, PTM-1250).

## ATAC-seq assay and data analysis

Three grams of 14-day-old seedlings were harvested, and nuclei were extracted. The nuclear suspension was incubated with Tn5 transposase in a transposition reaction mix at 37 °C for 30 min to fragment genomic DNA. The resulting DNA fragments were used for library preparation with the TruePrep DNA Library Prep Kit V2 for Illumina (Vazyme, TD501), following the manufacturer's protocol. Libraries were sequenced on an Illumina NovaSeq 6000 platform using 150-bp paired-end reads. For ATAC-seq data analysis, adapter sequences were trimmed from raw reads using fastp (v0.24.0). Cleaned reads were aligned to the rice reference genome (MSU7.0) using Bowtie2. Reads mapping to mitochondrial and chloroplast genomes were filtered to exclude organellar DNA contamination. Peak calling was performed with MACS2 (v2.2.9.1) using the following parameters: --nomodel --keep-dup all --shift -75 --extsize 150.

## Immunoprecipitation assay

14-days-old wild type seedlings were ground into power by liquid nitrogen, and nuclear proteins were extracted. For immunoprecipitation, nuclear lysates were incubated overnight at 4 °C with anti-H3K9bhb antibody (1:1000 dilution, PTM Bio, PTM-1250) or IgG (1:10000 dilution, Abclonal, AS070) conjugated to protein A magnetic beads (Invitrogen, 10001D). After four washes with PBST buffer (PBS containing 0.1% Tween-20; 10 min per wash at 4 °C), co-immunoprecipitated proteins were eluted in Laemmli buffer, resolved by SDS-PAGE, and analyzed by western blotting using the following primary antibodies: anti-H3 (Abcam, ab1791, 1:1000 dilution), anti-RPS6 (Abcam, ab40820, 1:1000 dilution), and anti-HSP70 (Abmart, M20041, 1:1000 dilution).

## Subcellular localization assay

The full-length cDNAs of RPS6, HSP70, and H3 were subcloned into the pCambia1301 vector. H3 was fused to RFP, while RPS6 and HSP70 were fused to GFP. *Agrobacterium tumefaciens* strains harboring these constructs were infiltrated into the abaxial epidermis of 6-week-old *N. benthamiana* leaves using a needleless syringe. After 48 h of incubation, subcellular localization was visualized using a laser-scanning confocal microscope (LSM980, Zeiss).

## RNA-seq data analysis

Total RNA from rice plants was isolated using TRIzol reagent (Invitrogen, 15596018). The Illumina TruSeq RNA Sample Preparation Kit was used to construct RNA libraries, which were then sequenced on the Illumina NovaSeq 6000 platform. For data analysis, FastP (v0.232) was employed to remove low-quality reads and adapters.

Subsequently, the cleaned reads were mapped to the rice genome (MSU 7.0) using FeatureCounts (version 2.0.3). DESeq2 (v1.36.0) was utilized to calculate genes with differential expression. Genes with a fold change >2 and a *P*-value <0.05 were considered significantly differentially expressed.

## In vitro de-Kbhb activity assay

The full-length coding sequences of *HDA705*, *SRT1*, and *SRT2* were individually ligated into the protein expression vector pET28a. Subsequently, the vectors were transformed into the *Escherichia coli* BL21 (DE3) strain to express the target proteins. HDA705-His, SRT1-His, and OsSRT2-His proteins were purified using HisSep Ni-NTA MagBeads (Qualityard, QYP1062). For the in vitro de-Kbhb activity assay, total proteins were extracted from rice leaves and then incubated with purified His-tagged histone deacetylases in a reaction buffer containing 50 mM Tris-HCl (pH 8.5), 2.7 mM KCl, 1 mM MgCl$_2$, 137 mM NaCl, and 1 mM dithiothreitol. The reaction mixture was incubated at 37 °C for 4 h. The final product was denatured at 95 °C and analyzed using immunoblotting with anti-Kbhb (1:1000 dilution, PTM, PTM-1201RM) and anti-H3K9bhb (1:1000 dilution, PTM, PTM-1250) antibodies.

## LC−MS/MS and data analysis

Rice flowers were ground into powder in liquid nitrogen and then lysed in lysis buffer (8 M urea, 1% Triton-100, 10 mM dithiothreitol, and 1% Protease Inhibitor Cocktail). The mixture was then sonicated three times in an ultrasonicator, followed by centrifugation at $13400 \times g$ at 4 °C for 10 min. Next, the supernatant was precipitated with 20% cold TCA for 2 h at −20 °C, followed by centrifugation at $13400 \times g$ at 4 °C for 10 min to obtain the protein precipitate. The final protein precipitate was washed three times with cold acetone. Subsequently, the protein was reduced in 5 mM dithiothreitol for 30 min at 56 °C and alkylated with 11 mM iodoacetamide for 15 min at room temperature in darkness. Then, the proteins were digested with trypsin overnight. After digestion, peptides were desalted with a C18 column (Millipore) and then dried in a vacuum. The peptides were then incubated with pre-washed antibody beads (PTM-1204, PTM Bio) in NETN buffer (100 mM NaCl, 1 mM EDTA, 50 mM Tris-HCl, 0.5% NP-40, pH 8.0) at 4 °C overnight with gentle shaking, and then eluted from the beads with 0.1% trifluoroacetic acid after three washes with NETN buffer. The Kbhb-enriched peptides were then vacuum-dried. Subsequently, the peptides were dissolved in solvent A (0.1% formic acid, 2% acetonitrile/ in water) and separated on a nanoElute UHPLC system (Bruker Daltonics) in solvent B (0.1% formic acid in acetonitrile) using a gradient from 6% to 22% over 40 min, followed by 22% to 30% in 4 min, and climbing to 80% in 10 min, all at a constant flow rate of 450 nL/min. The resulting peptides were analyzed by timsTOF Pro (Bruker Daltonics) mass spectrometry. The obtained mass spectra were searched against the rice Uniprot database using the MaxQuant search engine (v.1.6.15.0). The cleavage enzyme was set as Trypsin/P, allowing up to 2 missing cleavages. The mass tolerance for precursor ions was set as 20 ppm and 5 ppm in the first search and main search, respectively. Fragment ions of 0.02 Da were set for the mass tolerance. A fixed modification on Cys was specified as Carbamidomethyl, and protein N-terminal acetylation and Met oxidation were specified as variable modifications. An *FDR* value < 1% was adopted.

## Reporting summary

Further information on research design is available in the Nature Portfolio Reporting Summary linked to this article.

## Data availability

The mass spectrometry data produced by this research were deposited in the ProteomeXchange Consortium via the PRIDE partner repository with the dataset identifier PXD051126. The RNA-seq data, ChIP-seq data, and ATAC-seq data generated in this study were deposited to the

NCBI GEO database under accession codes GSE294824, GSE294828, and GSE294829. Other previously published RNA-seq data used in this study are available in the NCBI SRA database under accession codes: SRX1500162, SRX5197504, SRX5636373, SRX5636337 (SRX5636337), SRX1800518, SRX5636385. Previously published ATAC-seq and ChIP-seq data are under the accession codes: SRR10914733, SRR10751615, SRR10751619, SRX1044777, SRX189766, and SRX1620997. Source data are provided in this paper.

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

## Acknowledgements

This study was funded by the Guangxi Science Fund for Distinguished Young Scholars (2024JJG130008). Ba-Gui Youth Talent Support Program of Guangxi (to Qiutao Xu). The National Natural Science Foundation of China (32100465 and 32302302). The starting research grant for High-level Talents from Guangxi University (ZX01080033124005) and Scientific Research and Development Fund of the College of Agriculture, Guangxi University (EE101761).

## Author contributions

X.C. and Q.X. designed the experiments, X.C., Q.X., and Y.Z. wrote the paper. X.M. helped revised the paper. Q.X., Z.C., and R.W. performed the experiments. D.H., Y.D., and Q.X. performed the data analyses. J.C., Z.C., L.Z., J.H., and J.Z. provided technical support. All authors have read and approved the final manuscript.

## Competing interests

The authors declare no competing interests.
