## [Transparent Peer Review file. · Nature Communications]

Regulation of Plant Immunity Through Histone H3K9bhb-Mediated Transcriptional Control

Corresponding Author: Professor Xiaoyang Chen

Version 0:

Reviewer comments:

Reviewer #1

(Remarks to the Author)

In this study, the authors investigated the role of histone lysine β -hydroxybutyrylation in rice immunity. Kbhb is a novel PTM and it has not been investigated in plants before. First authors identified the site regions of Kbhb on plants, then applied exogenous β -hydroxybutyrate to induce H3K9bhb and investigated disease resistance. Finally, they performed Kbhb proteomic analysis to identify Kbhb sites and proteins. The Kbhb is novel in plants, however, there are problems with the manuscript.

1. It should be noted that whether protein/histone acetylation modification also plays an important role in plant disease. What is the relationship between histone H3K9bhb and H3K9ac modifications? Both modifications occur at the H3K9 site and use the same enzymes, histone acetyltransferases (HATs) and deacetylases (HDACs), are they competitive with each other?
2. There should be part of the relationship between PTM and plant immunity in the introduction section.
3. For the experimental treatment, how did the authors ensure that histone H3K9bhb but not H3K9ac happened on the plant immunity?
4. L162-165, H3K9ac plays a dominant role while H3K9bhb may play a complementary role, why do they investigate H3K9bhb but not H3K9ac for plant immunity?
5. In L189, the authors said that there was no change in H3K9ac levels, but from Fig 5a I can see that it was lower in the treatments than in the control.
6. L217, why was OsWAK90 chosen? The same question on three recombinant HDACs (SRT1, SRT2 and HDA705) in L274-277.
7. L250-251, what does a moderate positive correlation mean?
8. L289-293, HDA705 plays an important role in acetylation, how do the authors conclude that HDA705-mediated regulation of H3K9bhb plays a role in the plant-pathogen response? Doesn't acetylation play a role in plant immunity?
9. L357-359, more evidence is needed to support this point.
10. Plant immunity consists of pattern-triggered immunity (PTI) and effector-triggered immunity (ETI), what does the type of Kbhb induce? Which effectors might be involved?

Overall, the logical relationships are confusing, scientific issues are unclear and results do not support some of the conclusions

Reviewer #2

(Remarks to the Author)

Whereas protein hydroxybutyrylation has recently been reported in Arabidopsis, the submitted manuscript entitled 'Regulation of Plant Immunity Through Histone H3K9bhb-Mediated Transcriptional Control' is the first report of the existence of histone H3 hydroxybutyrylation in plants, and its behaviour as a typical chromatin mark associated with gene expression.

The manuscript is short, particularly the Discussion section, but very clear and easy to read. It reports a very large workload using multiple methodologies, with multiple ChIP-seq, RNAs-seq, plant pathogenic assays, LC-MS, and whole-cell mass spectrometry for the detection of plant hydroxybutyrylated proteins.

Focusing on rice, this work greatly benefited from a custom-made specific anti-histone H3 lysine β -hydroxybutyrylation (H3K9bhb) previously described in the landmark 2016 study 'Metabolic Regulation of Gene Expression by Histone Lysine β -

Hydroxybutyrylation', which first reported the existence of this histone mark in mammals. Equipped with this unique tool, the authors report the first plant H3K9bhb epigenomic landscape, its relationship with other chromatin hallmarks, including the related H3K9 acetylation, as well as H3K9bhb epigenomic variations in response to exogenous application of the H3K9bhb substrate (β -hydroxybutyrate) and of plant pathogens. This big dataset convincingly shows that H3K9bhb is globally enriched at chromatin in response to artificial BHB application, especially at genes contributing to rice immune responses to pathogens such as fungi. In turn, endogenous BHB is increased in response to various fungal infections and thereby triggers H3K9bhb deposition at multiple genes contributing to plant defense against fungi, which is demonstrated by looking at several typical hallmarks of plant defense (ROS production, sensitivity assays, etc). Last, the authors report several hundreds of rice proteins are hydroxybutyrylated in vivo, most notably plastid proteins including RuBisCo.

Altogether, this study is of good quality, with convincing results and interpretations, and reports important insights into chromatin modification and dynamics in response to biotic stress. Given the intricate links between protein hydroxybutyrylation and metabolism, the study will also be of great interest to the field of research in deciphering how metabolism impacts epigenome and plant-pathogen interactions.

Notwithstanding its quality, I have several suggestions and requests to confirm the main conclusions and for better addressing the key findings:

- No token was given to access and check the quality of the ChIP-seq and RNA-seq datasets. Confidential access to the processed normalized profiles (e.g. bigwig files) has to be given to the reviewers.
- The first Results section focuses on H3K9 hydroxybutyrylation being frequently found at the same genes than H3K9 acetylation. This suggests but does not directly address the possibility that hydroxybutyrylation competes with acetylation, or relies on prior acetylation, or whether both marks could be found simultaneously on the same chromatin fragments. In my opinion this is the major caveat of this study, and should be addressed by doing
 - 1) bioinformatics analyses testing whether H3K9ac is usually reduced at loci gaining H3K9bhb in response to BHB application, or other analyses testing whether any sign of competition can be detected in the rice epigenome dynamics,
 - 2) performing a set of sequential ChIP (Re-ChIP) combining anti-H3K9ac and anti-H3K9bhb ChIP, in both directions and with proper controls using only one antibody at a time. This would address whether H3K9ac and H3K9bhb can be found on the same nucleosomes (yet on different histones H3), on the same chromatin fragments, or replace each other. This would clarify the discussion about both marks being frequently found on the same genes, and maybe also test whether H3K9ac is a pre-requisite for H3K9bhb deposition.
- The latter issue can be addressed using H3K9bhb western-blot and ChIP analyses of mutants with reduced H3K9 acetylation as well as in SRT2 and HDA705 mutant plants.
- H3K9bhb was found to impact immune response genes, and accordingly, BHB exogenous application reduces plant sensitivity to fungal infections. Yet, no MEME promoter sequence analyses of H3K9bhb-marked genes are performed. This should be done to address whether some sequence motifs and transcription factors are linked to H3K9bhb gene specificity.
- In Figure 5d and along the manuscript. In the absence of exogenous reference (spike-in), the quantification of H3K9bhb global increase at chromatin by ChIP-seq can be strongly underestimated. Hence, at minima, the number of H3K9bhb-marked genes should be compared in the different conditions tested where the H3K9bhb global level is changed. For example, are there more marked genes, or do genes gain even more H3K9bhb in response to BHB application or to fungus infection?
- Figure 5i must be completed by a scatter-plot comparing H3K9bhb ChIP-seq and RNAseq datasets to test whether both chromatin and expression changes correlate at all or a subset of genes.
- Endogenous production of BHB is nicely quantified by LC-MS/MS. Where is it produced in the plants, and in which subcellular compartment? is there any detectable BHB in the nucleus?
- Why the OsWAK90 gene was selected for a functional assay with CRISPR/Cas9? What is known about this gene, how is it regulated, and is it targeted by SRT2 and HDA705?
- Figure 7d, as for Figure 5d, we lack understanding of the number of H3K9bhb marked genes in the HDA705 and SRT2 mutant plants and compare them to the marked genes in WT plants to assess the real impact of these two HDACs on H3K9bhb erasure. Meta-gene profiles of H3K9bhb at the whole set of genes, and at the subset of genes gaining/losing completely the mark, should also be given with proper normalization.
- The mass spec detection of protein hydroxybutyrylation will represent an important public resource. Yet, about chromatin control, no information is given on the hydroxybutyrylation of histones other than H3. To connect this part of the manuscript with the rest of the study, it appears necessary to give information on all histones hydroxybutyrylation, including which histone H3 variants are hydroxybutyrylated, and also which known chromatin modifiers and remodelers, including all known HAT and HDACs that could serve in histone PTM regulatory loops.
- The Discussion does not address the critical point of how hydroxybutyrylation could affect gene expression. Can hydroxybutyrylation affect histone H3 affinity for DNA, as acetylation does?

- Also, is H3K9bhb expected to affect transcription through chromatin accessibility? If so, this could be tested using ATAC-seq or MNase-seq analyses upon BHB application.

Reviewer #3

(Remarks to the Author)

Lysine β -hydroxybutyrylation (Kbhb) is a new type of histone mark, while its prevalence and function remain unclear in plants. Xu et al. performed a genome-wide profiling of histone H3K9bhb in rice, and found histone H3K9bhb marks were highly enriched at transcription start sites (TSSs), which positively correlated with the active histone marks H3K4ac, H3K9ac, and H3K4me3. Further ChIP-seq data showed that rice H3K9bhb modification was enriched at the genes involved in the defense response to fungal infection. Increasing H3K9bhb modification in rice, results in higher resistant on *U. virens* infection, suggesting H3K9bhb may have a key role in plant defense mechanisms. The authors also found that SRT2 and HDA705 had notable de-Kbhb activity in rice. At last, they also performed a proteome-wide identification of Kbhb substrates in rice flowers, and found Kbhb modified proteins play important roles in plant metabolism.

The overall story is interesting, which includes huge amount of data and disclosed the distribution, deposition, and function of histone Kbhb in rice. However, this manuscript spends most of the space to focus on the omics data, failed to reveal the fundamental mechanisms of Kbhb modification in rice. Although there is evidence that Kbhb help to increase fungal infection in rice, whereas it lacks the direct evidence to support the function of Kbhb modification in rice immune system. HDACs have de-Kbhb activity, while there is no evidence that showed SRT2 and HDA705 have roles in response to fungal infection. To prove the direct writer and eraser of histone Kbhb in rice, it needs more evidence that should include *in vivo*, *in vitro* and genetic experiments, and the current evidence is not sufficient. The last part "Proteome-wide identification of Kbhb substrates in rice flowers" seems to stand alone and does not effectively contribute to the main points of the paper. It appears that these data points are not well - integrated into the overall argument, and as a result, they do not enhance the validity or persuasiveness of your research.

All in one, the current manuscript is not suitable for publication in Nature Communications, however, it is worth a re-submit opportunity to take a step back and carefully assess the overall logic and essential evidence showed in this manuscript.

Reviewer #4

(Remarks to the Author)

Qiutao et al. investigate the presence of H3K9bhb as an epigenetic mark in rice and its role in plant defenses. They discovered that H3K9bhb is associated with actively transcribed genes and identified functional differences between this mark and H3K9ac. Interestingly, H3K9bhb is enriched in genes associated with plant defenses. They also demonstrate that exogenous BHB application induces H3K9bhb accumulation and enhances plant defenses, while plant infection triggers endogenous accumulation of this mark in defense genes. Furthermore, they reveal that OsSRT2 and OsHDA705 can remove this epigenetic mark.

I found the manuscript highly interesting and easy to read, providing valuable insights for further research. However, I have a major concern that needs to be addressed. The authors present enough evidence of the existence and/or presence of H3K9bhb in rice, but an assay demonstrating the *in vivo* specificity of the antibody would be better. While Figure S1 shows the antibody's specificity *in vitro*, comparing its cross-reactivity with H3K14bhb, H3K9cr, H3K9la, and H3K9ac, I remain concerned about potential cross-reactivity with other proteins, particularly those in the nucleus, where 15% of the proteins immunoprecipitated with anti-Kbhb are located (Figure 8). This suggests that other proteins beyond those tested in Figure S1 could also be targeted in ChIP assays or western blots, as shown in Figure 5.

I do not question the broad conclusions, but without proper endogenous controls, the presence of this epigenetic mark in the genome could be overestimated or biased (especially since many of the IPed proteins related to plant-pathogen interactions, as shown in Figure 8f, could skew the results). Could the authors perform a WB with anti-H3K9bhb, followed by LC-MS/MS of the purified band corresponding to H3K9bhb, probably in BHB-treated and untreated plants? I would expect an enrichment of H3K9 peptides with bhb PTM rather than other PTMs or unrelated proteins. Alternatively, overexpressing HDA705 or SRT2 to reduce H3K9bhb levels and use ChIP-qPCR to assess H3K9bhb presence on specific target genes? While this approach might be biased if these HDACs also de-Kbhb other cross-reacting proteins, it would help address this concern.

Minor Concerns:

Could the authors clarify Figure 2C? Why does the peak of reads (from H3K9bhb ChIP-seq) appear near the TTS, unlike Figures 1G and 2A, where peaks are mainly near the TSS?

The authors present three clusters of H3K9bhb-marked genes alongside other epigenetic marks and analyze GO terms for cluster 1, showing an enrichment of stress-induced genes among others. What about the genes in the other two clusters? Do they differ from those in cluster 1? This could hint at distinct roles for H3K9bhb in combination with different epigenetic marks.

I suggest presenting the results in Figure 3 before those in Figure 2. Figure 3 provides a broader context (e.g., GO analysis and correlation with H3K9ac), while Figure 2 distinguishes between different gene classes marked by H3K9bhb. This is a suggestion for the authors to consider.

The authors show that SRT2 and HDA705 regulate H3K9bhb levels in pathogen-responsive genes. The ChIP-seq analysis

in hda705 mutant plants and the in vitro de-Kbhb activity assay suggest that HDA705 may have de-Kbhb activity. To strengthen this point, I suggest immunoprecipitating HDA705 to test its association with H3K9bhb-enriched sequences in vivo via ChIP-qPCR on specific targets.

The statistical analysis in Figure 3C is missing.

Line 116: "genes (d)" should be corrected to "genes (Fig. 1d)."

Version 1:

Reviewer comments:

Reviewer #1

(Remarks to the Author)

Authors have resolved all my concerns, it can be accepted now.

Reviewer #2

(Remarks to the Author)

The revised version of the manuscript incorporates a significant amount of additional workload, including the requested analyses of H3K9ac/H3K9bhb relationships through sequential ChIP (Re-ChIP), and the apparent influence of H3K9 hydroxybutyrylation on gene expression by modulating chromatin accessibility.

In my opinion the manuscript is suitable for publication in Nature Coms provided that:

1) the conclusion that H3K9ac and H3K9bhb are two mutually exclusive histone H3K9 modifications is better argued in the Discussion by compiling all relevant evidence from this study - not just Re-ChIP-qPCR analysis of a few selected genes.

2) this conclusion is used in all instances for accurate data interpretation, hence proposals that genes are dually marked by H3K9ac and H3K9bhb (e.g. lane 144) should be re-visited or more cautiously formulated. As discussed, apparent dual marking of a given gene could result from individual marking by each of these two modifications in different cells.

3) the ATAC-seq data are better integrated with gene expression (RNA-seq) and chromatin profiling (H3K9bhb ChIP-seq) to sustain the possibility that H3K9bhb is linked to increased DNA accessibility and gene expression control at specific immunity gene targets.

Reviewer #3

(Remarks to the Author)

The current version is much better, while there are still some concerns about logic problems in this manuscript.

1. For the second part "Functional divergence of H3K9bhb and H3K9ac" (Line 142), why the authors want to investigate the correlation between H3K9bhb and H3K9ac? Are there any evidence to show that H3K9bhb functions together with H3K9ac? There should have the transitional phrases to explain why the authors want to start the study of this part. Similar problems are shown in the third part "Comparative analysis of H3K9bhb, histone H3 acetylation, and histone H3 methylation modification" (Line 172).

2. Are there any rice mutants with null or low level of β -hydroxybutyrylation? If lower down the total level of β -hydroxybutyrylation in vivo, what are the development or disease resistant phenotypes of these mutants?

If the authors could fully address the two questions as described above, I think the manuscript would be ready for acceptance by Nature Communications.

Reviewer #4

(Remarks to the Author)

The authors have included sufficient results to address my concerns. In addition, the revisions made in response to the other reviewers' comments have further improved the manuscript. Overall, I find the work very interesting and valuable.

Reviewer's Comments:

Reviewer #1 (Remarks to the Author):

In this study, the authors investigated the role of histone lysine β -hydroxybutyrylation in rice immunity. Kbhb is a novel PTM and it has not been investigated in plants before. First authors identified the site regions of Kbhb on plants, then applied exogenous β -hydroxybutyrate to induce H3K9bhb and investigated disease resistance. Finally, they performed Kbhb proteomic analysis to identify Kbhb sites and proteins. The Kbhb is novel in plants, however, there are problems with the manuscript.

1. It should be noted that whether protein/histone acetylation modification also plays an important role in plant disease. What is the relationship between histone H3K9bhb and H3K9ac modifications? Both modifications occur at the H3K9 site and use the same enzymes, histone acetyltransferases (HATs) and deacetylases (HDACs), are they competitive with each other?

Response:

Yes, we acknowledge that histone acetylation plays a crucial role in plant immunity. We have mentioned this in our introduction part (Lines 57-66).

Regarding the relationship between H3K9bhb and H3K9ac, our findings suggest that these modifications are competitive. This competition was confirmed by our ChIP-re-ChIP-qPCR results in this revision (Supplementary Fig. 7), which indicate mutual exclusivity at the same loci. However, despite their shared location, these modifications have distinct functional roles. GO analysis revealed that H3K9ac-marked genes are predominantly enriched in developmental processes, whereas H3K9bhb-marked genes are associated with stress response pathways, particularly immunity-related responses (Fig. 2d). This functional divergence was further supported by our H3K9ac (obtained in this revision) and H3K9bhb ChIP-seq data in the *hda705* mutant, which showed minimal overlap between genes with increased H3K9ac and those with increased H3K9bhb (Supplementary Fig. 23c).

In summary, while H3K9bhb and H3K9ac compete for occupancy at the same histone loci, they regulate distinct biological processes, with H3K9ac primarily involved in development and H3K9bhb in stress and immunity responses.

2. There should be part of the relationship between PTM and plant immunity in the introduction section.

Response:

Thank you for the suggestion. We have added a paragraph in the introduction to discuss the relationship between post-translational modifications (PTMs) and plant immunity. Please see Lines 57-70.

3. For the experimental treatment, how did the authors ensure that histone H3K9bhb but not H3K9ac happened on the plant immunity?

Response:

We examined both H3K9ac and H3K9bhb levels under *U. virens* infection and found that the

infection specifically led to an increase in H3K9bhb, while H3K9ac levels remained unchanged (Fig. 6b). Additionally, we analyzed the H3K9ac levels of the same genes that exhibited increased H3K9bhb upon infection and observed no significant difference between *U. virens*-infected and control (CK) samples (Supplementary Fig. 14 and 15). Furthermore, in *hda705* mutants, immunity-related genes that showed elevated H3K9bhb levels (Fig. 7g) did not exhibit any significant changes in H3K9ac levels (Supplementary Fig. 22). Collectively, these results indicate that H3K9bhb plays a crucial role in rice immune responses.

4. L162-165, H3K9ac plays a dominant role while H3K9bhb may play a complementary role, why do they investigate H3K9bhb but not H3K9ac for plant immunity?

Response:

Yes, our data suggest that H3K9ac plays a dominant role in gene regulation, while H3K9bhb may serve a complementary function. However, GO analysis indicates that H3K9ac-marked genes are primarily enriched in developmental processes, whereas H3K9bhb-marked genes are more associated with stress response pathways, including plant immunity (Fig. 2d). Additionally, our Western blot (Fig. 6b) and ChIP-qPCR (Supplementary Fig. 14 and 15) results showed that *U. virens* infection specifically induced H3K9bhb, with no obvious changes in H3K9ac levels. These findings suggest that H3K9bhb plays a key role in plant immunity, which is why we focused on investigating its function rather than H3K9ac.

5. In L189, the authors said that there was no change in H3K9ac levels, but from Fig 5a I can see that it was lower in the treatments than in the control.

Response:

Thanks for pointing this out, we have revised our claim in the MS (Lines 215-216).

6. L217, why was OsWAK90 chosen? The same question on three recombinant HDACs (SRT1, SRT2 and HDA705) in L274-277.

Response:

Based on previous study showing that wall-associated kinases (WAKs) act as positive regulators of fungal disease resistance in various plant species (Delteil et al., 2016), and given that *OsWAK90* is strongly induced upon pathogen infection (Delteil et al., 2016), we selected this gene for further investigation, as it may play a key role in disease resistance. We have revised our statement in MS for clarity (Lines 258-260).

For the *in vitro* enzyme activity assay of SRT1, SRT2, and HDA705, our selection was based on immunoblot analysis of *hdac* mutants, which revealed that only *srt1*, *srt2*, and *hda705* mutants exhibited increased H3K9bhb levels (Figure 7a). This suggests that these three genes may possess de-Kbhb activity, warranting their inclusion in the *in vitro* enzyme activity assay. We have revised our statement accordingly for clarity (Lines 315-317).

Reference

Delteil, A., Gobbato, E., Cayrol, B., Estevan, J., Michel-Romiti, C., Dievart, A., Kroj, T. & Morel, J. B. Several wall-associated kinases participate positively and negatively in basal defense against rice blast fungus. *BMC Plant Biol.* **16**, 1-10 (2016)

7. L250-251, what does a moderate positive correlation mean?

Response:

A moderate positive correlation means that there is a noticeable but not strong relationship between two variables, where as one variable increases, the other tends to increase as well. In statistical terms, this usually corresponds to a correlation coefficient (r) between 0.3 and 0.7. We have added the explanation in the MS (Line 291).

8. L289-293, HDA705 plays an important role in acetylation, how do the authors conclude that HDA705-mediated regulation of H3K9bhb plays a role in the plant-pathogen response? Doesn't acetylation play a role in plant immunity?

Response:

To further investigate the role of HDA705, we performed ChIP-seq analysis of H3K9ac in *hda705* mutants and WT plants in this revision. The results revealed that the loss of *HDA705* led to an increase in H3K9ac levels in 2,258 genes, which were primarily enriched in pathways related to diterpenoid metabolism, gibberellin metabolic processes, response to sucrose, organic anion transport, and response to monosaccharides, among others (Supplementary Fig. 23a, b). However, no enrichment was observed in pathways associated with plant immunity. In contrast, genes with increased H3K9bhb levels in *hda705* mutants were enriched in various pathways, including those associated with plant immunity (Fig. 7f). These findings indicate that HDA705-mediated regulation of H3K9bhb plays a specific role in the plant – pathogen response, distinct from its role in H3K9ac-mediated gene regulation.

9. L357-359, more evidence is needed to support this point.

Response:

Thanks for the suggestion. Lines 357 – 359 have been moved to Lines 468 – 469 in this revision. To further support our conclusion, we examined the levels of H3K9ac and H3K9bhb under *U. virens* infection and found that the infection specifically induced an increase in H3K9bhb levels, while H3K9ac levels remained unchanged (Fig. 6b). Additionally, we assessed the H3K9ac levels of the same genes that exhibited increased H3K9bhb under *U. virens* infection and observed no significant differences between infected and CK samples (Supplementary Fig. 14 and 15). Furthermore, immunity-related genes that showed increased H3K9bhb in *hda705* mutants exhibited no detectable changes in H3K9ac levels (Fig 7g, Supplementary Fig. 22). To gain further insight, we performed ChIP-seq analysis of H3K9ac in *hda705* mutants and WT plants. The analysis revealed minimal overlap between genes with increased H3K9bhb and those with increased H3K9ac in *hda705* mutants (Supplementary Fig. 23c). Moreover, HDA705-regulated H3K9ac genes were primarily involved in metabolic processes and development (Supplementary Fig. 23b), whereas genes with increased H3K9bhb were enriched in pathways associated with plant immunity (Fig 7f). We have revised the discussion by integrating the above information (Lines 465 – 468). These findings provide additional evidence supporting the claim made in Lines 468 – 469.

10. Plant immunity consists of pattern-triggered immunity (PTI) and effector-triggered immunity

(ETI), what does the type of Kbh_b induce? Which effectors might be involved?

Response:

In fact, we are currently unable to determine whether the observed effects are specific to the PTI or ETI immune pathways, and we speculate that both may be involved. This speculation is supported by our Kbh_b mass spectrometry data, which revealed that several RLK (Receptor-Like Kinase) and NLR (Nucleotide-Binding Leucine-Rich Repeat) proteins—key components of PTI and ETI, respectively—are modified by Kbh_b.

Our previous study demonstrated that Uv1809 as a key effector targeting the rice histone deacetylase OsSRT2, Uv1809 increases the deacetylation activity of OsSRT2 (Chen et al., 2024). Following protein extraction from 35S-Uv1809 and 35S-EV rice leaves, immunoblotting analysis demonstrated reduced the levels of H3K9bh_b in 35S-Uv1809 compared to 35S-EV rice plants (Figure 1 in this document), indicating that the fungal effector Uv1809 from *U. virens* might target OsSRT2 to mediate plant H3K9bh_b level. Therefore, effectors targeting the writers, erasers or readers, it is possible to regulate the modification of plant histone Kbh_b.

Figure 1. Analysis of H3K9bh_b levels in 35S-Uv1809 and 35S-EV rice plants. Anti-H3 was used as control.

Reference

Chen, X. et al. *Ustilagoidea virens*-secreted effector Uv1809 suppresses rice immunity by enhancing OsSRT2-mediated histone deacetylation. *Plant Biotechnol. J.* **22**, 148–164 (2024).

Reviewer #2 (Remarks to the Author):

Whereas protein hydroxybutyrylation has recently been reported in Arabidopsis, the submitted manuscript entitled ‘Regulation of Plant Immunity Through Histone H3K9bh_b-Mediated Transcriptional Control’ is the first report of the existence of histone H3 hydroxybutyrylation in plants, and its behaviour as a typical chromatin mark associated with gene expression.

The manuscript is short, particularly the Discussion section, but very clear and easy to read. It reports a very large workload using multiple methodologies, with multiple ChIP-seq, RNAs-seq, plant pathogenic assays, LC-MS, and whole-cell mass spectrometry for the detection of plant hydroxybutyrylated proteins.

Focusing on rice, this work greatly benefited from a custom-made specific anti-histone H3 lysine β -hydroxybutyrylation (H3K9bh_b) previously described in the landmark 2016 study ‘Metabolic Regulation of Gene Expression by Histone Lysine β -Hydroxybutyrylation’, which first reported the existence of this histone mark in mammals. Equipped with this unique tool, the authors report the

first plant H3K9bhb epigenomic landscape, its relationship with other chromatin hallmarks, including the related H3K9 acetylation, as well as H3K9bhb epigenomic variations in response to exogenous application of the H3K9bhb substrate (β -hydroxybutyrate) and of plant pathogens. This big dataset convincingly shows that H3K9bhb is globally enriched at chromatin in response to artificial BHB application, especially at genes contributing to rice immune responses to pathogens such as fungi. In turn, endogenous BHB is increased in response to various fungal infections and thereby triggers H3K9bhb deposition at multiple genes contributing to plant defense against fungi, which is demonstrated by looking at several typical hallmarks of plant defense (ROS production, sensitivity assays, etc). Last, the authors report several hundreds of rice proteins are hydroxybutyrylated in vivo, most notably plastid proteins including RuBisCo.

Altogether, this study is of good quality, with convincing results and interpretations, and reports important insights into chromatin modification and dynamics in response to biotic stress. Given the intricate links between protein hydroxybutyrylation and metabolism, the study will also be of great interest to the field of research in deciphering how metabolism impacts epigenome and plant-pathogen interactions.

Notwithstanding its quality, I have several suggestions and requests to confirm the main conclusions and for better addressing the key findings:

No token was given to access and check the quality of the ChIP-seq and RNA-seq datasets. Confidential access to the processed normalized profiles (e.g. bigwig files) has to be given to the reviewers.

Response:

Thank you for the suggestion. We have deposited the ChIP-seq and RNA-seq datasets into the GEO database, and also provided the BigWig files for visualization. In addition, the access tokens have been included (Lines 740-742).

- The first Results section focuses on H3K9 hydroxybutyrylation being frequently found at the same genes than H3K9 acetylation. This suggests but does not directly addresses the possibility that hydroxybutyrylation competes with acetylation, or relies on prior acetylation, or whether both marks could be found simultaneously on the same chromatin fragments. In my opinion this is the major caveat of this study, and should be addressed by doing

1) bioinformatics analyses testing whether H3K9ac is usually reduced at loci gaining H3K9bhb in response to BHB application, or other analyses testing whether any sign of competition can be detected in the rice epigenome dynamics.

Response:

Thanks for the suggestion.

Since we do not have H3K9ac ChIP-seq data under BHB treatment, but have generated H3K9ac ChIP-seq data for *hda705* in this revision, we analyzed the H3K9ac and H3K9bhb datasets to test whether H3K9ac is typically reduced at loci gaining H3K9bhb, given *hda705*'s established role in modulating both modifications, as demonstrated by the results presented here (Figure 7, Supplementary Fig. 23) and in our previous study (Lu et al., 2018). In total, we identified 24,147

genes with quantitative fold changes in both H3K9ac and H3K9bhb datasets in *hda705* compared to wild type. Among these, nearly 60% of the genes exhibited opposite dynamic changes (i.e., increased H3K9ac with decreased H3K9bhb, or vice versa), and a moderate negative correlation ($r = -0.3$) was observed between these changes. Collectively, these datasets support the notion that H3K9ac is typically reduced at loci gaining H3K9bhb. We have integrated this result into the manuscript (Supplementary Fig. 23d).

Reference

Lu, Y., Xu, Q., Liu, Y., Yu, Y., Cheng, Z., Zhao, Y. & Zhou, D. Dynamics and functional interplay of histone lysine butyrylation, crotonylation, and acetylation in rice under starvation and submergence. *Genome Biol.* **19**, 144 (2018).

2) performing a set of sequential ChIP (Re-ChIP) combining anti-H3K9ac and anti-H3K9bhb ChIP, in both directions and with proper controls using only one antibody at a time. This would address whether H3K9ac and H3K9bhb can be found on the same nucleosomes (yet on different histones H3), on the same chromatin fragments, or replace each other. This would clarify the discussion about both marks being frequently found on the same genes, and maybe also test whether H3K9ac is a pre-requisite for H3K9bhb deposition.

Response:

To investigate whether H3K9ac competes with H3K9bhb at the same loci, we designed primers targeting regions where both H3K9ac and H3K9bhb peaks were detected. We then performed H3K9bhb ChIP, followed by H3K9ac re-ChIP-qPCR. Among the six genes analyzed (including one negative control from a no-peak region), all exhibited a mutually exclusive pattern—when the H3K9 site was occupied by Kbhb modification, no detectable Kac signal was observed. This result has been integrated into Supplementary Fig. 7b. These results suggest a competitive relationship between Kbhb and Kac at the H3K9 position.

To further strengthen this conclusion, and as recommended by the reviewer, we performed a reverse ChIP-re-ChIP experiment—first conducting H3K9ac ChIP, followed by H3K9bhb re-ChIP-qPCR. H3K4ac was used as a control. Among the six genes analyzed (including one negative control from a no-peak region), all showed that when H3K9 was occupied by Kac modification, no detectable Kbhb signal was present. This result has been integrated into Supplementary Fig. 7c. Collectively, these results confirm that H3K9ac and H3K9bhb compete for occupancy at the H3K9 position.

Based on our new findings that H3K9ac and H3K9bhb compete for the same genomic sites (Supplementary Fig. 7), and that a large number of genes are exclusively marked by H3K9bhb (Fig. 2a), along with the observation that pathogen infection increases H3K9bhb levels without affecting H3K9ac (Fig. 6b), it is reasonable to conclude that H3K9ac is not a prerequisite for H3K9bhb deposition.

- The latter issue can be addressed using H3K9bhb western-blot and ChIP analyses of mutants with reduced H3K9 acetylation as well as in SRT2 and HDA705 mutant plants.

Response:

We appreciate this insightful suggestion. To systematically examine the regulatory interplay between H3K9ac and H3K9bhb, we focused on the *hda705* mutant, given its established role in

modulating both modifications. Our immunoblot analysis revealed significantly elevated H3K9bhb levels in *hda705* mutants in this study (Fig. 7a), while our previous work demonstrated a parallel increase in H3K9ac in the same genetic background (Lu et al., 2018). Since these two modifications compete for the same genomic regions, it is plausible that HDA705-regulated H3K9ac and H3K9bhb may have distinct functional outcomes. This is supported by the limited overlap between genes with increased H3K9bhb and those with increased H3K9ac (Supplementary Fig. 23c).

In addition, among the 24,147 genes identified with quantitative fold changes in both the H3K9ac and H3K9bhb datasets in *hda705*, we observed that more than half of the genes (see response to question “1”) exhibited opposite dynamic changes (i.e., increased H3K9ac with decreased H3K9bhb, or vice versa), and a moderate negative correlation ($r = -0.3$) was found between these changes (Supplementary Fig. 23d).

Based on these epigenomic dynamics (Supplementary Fig. 23d) and ChIP-reChIP-qPCR (Supplementary Fig. 7) results, we conclude that there is a competitive relationship between Kbbh and Kac at the H3K9 position.

Reference

Lu, Y., Xu, Q., Liu, Y., Yu, Y., Cheng, Z., Zhao, Y. & Zhou, D. Dynamics and functional interplay of histone lysine butyrylation, crotonylation, and acetylation in rice under starvation and submergence. *Genome Biol.* **19**, 144 (2018).

- H3K9bhb was found to impact immune response genes, and accordingly, BHB exogenous application reduces plant sensitivity to fungal infections. Yet, no MEME promoter sequence analyses of H3K9bhb-marked genes are performed. This should be done to address whether some sequence motifs and transcription factors are linked to H3K9bhb gene specificity.

Response:

We have conducted a motif analysis of the promoters of H3K9bhb-marked genes. The analysis revealed 53 motifs, including known transcription factor motifs such as bZIP910, NAC096, WRKY29, and MYB98. These results suggest that these transcription factors may specifically regulate the expression of H3K9bhb-marked genes. These findings have been integrated into the manuscript (Lines 133-136, Supplementary Fig. 5) and further discussed in the discussion section (Lines 484-486).

- In Figure 5d and along the manuscript. In the absence of exogenous reference (spike-in), the quantification of H3K9bhb global increase at chromatin by ChIP-seq can be strongly underestimated. Hence, at minima, the number of H3K9bhb-marked genes should be compared in the different conditions tested where the H3K9bhb global level is changed. For example, are there more marked genes, or do genes gain even more H3K9bhb in response to BHB application or to fungus infection?

Response:

Thanks for the suggestion. Regarding the spike-in, we did not use it in our data analysis. One reason for this was that the normalization method we used in DiffBind requires the raw read counts as input,

and spike-ins are not necessary for this approach. We agree that with the spike-ins approach it would be better for such data analysis. Unfortunately, spike-ins controls were not included in the samples for high throughput sequencing. However, we have addressed this issue by performing ChIP-qPCRs to validate the ChIP-seq results (Please see Fig. 7g, Supplementary Fig. 6, 12, and 14).

In comparing the number of H3K9bhb-marked genes under BHB application and fungal infection, differential analysis revealed that BHB application resulted in a greater number of hyper-H3K9bhb genes (6,766) compared to fungal infection (3,420) (Fig. 2a in this document). Cumulative density plots of fold changes demonstrated a significant increase in the BHB application group relative to the fungal infection group (Fig. 2b in this document). Further analysis showed that 89% of the genes exhibited higher fold changes in response to BHB application compared to fungal infection (Fig. 2c in this document). Collectively, these results indicate that more H3K9bhb was gained with BHB application than with fungal infection.

Figure 2. Analysis of fold changes in H3K9bhb-upregulated genes under BHB application and fungal infection. (a) Comparison of the number of H3K9bhb-upregulated genes under BHB application and fungal infection. (b) Cumulative density plots of fold changes in H3K9bhb-upregulated genes under BHB application and fungal infection. (c) Heatmap of fold changes in H3K9bhb-upregulated genes under BHB application and fungal infection.

- Figure 5i must be completed by a scatter-plot comparing H3K9bhb ChIP-seq and RNAseq datasets to test whether both chromatin and expression changes correlate at all or a subset of genes.

Response:

Thanks for the suggestion. We have replaced the Venn diagram with a scatter plot. Our analysis showed a relatively low positive correlation ($r = 0.11$) between the fold changes of ChIP-seq and RNA-seq datasets. Additionally, we observed a relatively strong correlation ($r = 0.42$) within a subset of genes ($N = 1100$, Supplementary Fig. 11). This result has been integrated into the MS (Lines 238-243).

- Endogenous production of BHB is nicely quantified by LC-MS/MS. Where is it produced in the plants, and in which subcellular compartment? is there any detectable BHB in the nucleus?

Response:

In animals, β -hydroxybutyrylation is closely associated with ketone body metabolism, particularly during carbohydrate deprivation or fasting, when β -hydroxybutyrate accumulates as an alternative energy source. While the source of β -hydroxybutyrate in plants remains unclear, evidence suggests that Kbhb may originate from fatty acid metabolism, as thousands of Kbhb sites have been identified

on proteins in lipid-rich algae (Ouyang et al., 2024). Here, we measured the content of BHB in the cytoplasm and nucleus using LC-MS/MS and confirmed its presence within the plant nucleus (Lines 115-117 Supplementary Fig. 1a). In addition, we found that exogenous application of BHB significantly increased its content in rice spikelets. These results have been incorporated into the manuscript (Lines 211-213, Supplementary Fig. 1b).

Reference

Ouyang, L., You, W., Poetsch, A., and Wei, L. Global Profiling of Protein Phosphorylation, Acetylation, and beta-Hydroxybutyrylation in *Nannochloropsis oceanica*. *J. Agric. Food Chem* **72**, 26248-26262 (2024).

- Why the *OsWAK90* gene was selected for a functional assay with CRISPR/Cas9? What is known about this gene, how is it regulated, and is it targeted by SRT2 and HDA705?

Response:

Previous study has shown that wall-associated kinases (WAKs) are involved in disease responses, as WAK genes are often regulated during bacterial infection in plants (Delteil et., 2016). During early infection stages (2 and 4 hours post-inoculation), the expression of *OsWAK90* was induced (Delteil et., 2016), suggesting that it may act as a positive regulator of plant pathogen resistance. Therefore, we selected this gene for functional analysis using CRISPR/Cas9. We have revised our manuscript to clarify this point (Lines 258 – 260).

According to the H3K9bhb ChIP-seq results for *srt2* and *hda705*, *OsWAK90* does not appear to be directly regulated by either of these histone deacetylases. Further study is needed to identify its regulator.

Reference

Delteil, A., Gobbato, E., Cayrol, B., Estevan, J., Michel-Romiti, C., Dievart, A., Kroj, T. & Morel, J. B. Several wall-associated kinases participate positively and negatively in basal defense against rice blast fungus. *BMC Plant Biol.* **16**, 1-10 (2016).

- Figure 7d, as for Figure 5d, we lack understanding of the number of H3K9bhb marked genes in the HDA705 and SRT2 mutant plants and compare them to the marked genes in WT plants to assess the real impact of these two HDACs on H3K9bhb erasure. Meta-gene profiles of H3K9bhb at the whole set of genes, and at the subset of genes gaining/losing completely the mark, should also be given with proper normalization.

Response:

Thank you for the suggestion. In response, we have included the number of H3K9bhb-marked genes in the *hda705* mutant and wild type plants. A total of 30,267 H3K9bhb-marked genes were identified in WT plants, while 31,703 were detected in *hda705* mutants. Among them, 28,746 genes were commonly marked in both genotypes. Compared with WT, *hda705* exhibited a gain of 2,957 and a loss of 1,521 H3K9bhb-marked genes (Lines 328-332, Supplementary Fig. 20).

Regarding the *srt2* mutant, we had not previously analyzed the dynamic changes of H3K9bhb. As suggested, we have performed H3K9bhb ChIP-seq in the *srt2* mutant and WT plants in this revision. Our analysis showed that a total of 30,863 H3K9bhb-marked genes were identified in WT plants,

while 31,210 were detected in *srt2* mutants. Among them, 27,982 genes were commonly marked in both genotypes (Lines 367-372, Supplementary Fig. 24b). Compared with WT, *srt2* exhibited a gain of 3,228 and a loss of 2881 H3K9bhb-marked genes (Supplementary Fig. 24b).

For normalization, we applied DESeq2 size factor normalization, a method used in DiffBind that adjusts for variations in sequencing depth and library size. This approach computes size factors for each sample, which are then used to scale the read counts, ensuring that the differences in H3K9bhb modifications are accurately represented across the samples. We have added the normalization method we used in all meta-gene plots (Lines 1023-1024, Lines 1026-1027, Lines 1039-1040, Lines 1062-1063, Lines 1065-1066, Lines 1079-1080, Lines 1093-1094, Lines 1096-1097).

- The mass spec detection of protein hydroxybutyrylation will represent an important public resource. Yet, about chromatin control, no information is given on the hydroxybutyrylation of histones other than H3. To connect this part of the manuscript with the rest of the study, it appears necessary to give information on all histones hydroxybutyrylation, including which histone H3 variants are hydroxybutyrylated, and also which known chromatin modifiers and remodelers, including all known HAT and HDACs that could serve in histone PTM regulatory loops.

Response:

Thanks for the suggestion. We have provided the histone sites that are hydroxybutyrylated. In total, our data identified 26 histone Kbhb sites, including several Kbhb sites on histones H3 and H4, as well as multiple Kbhb sites on histones H2A and H2B. Additionally, 20 transcription factors and five chromatin regulators were identified as Kbhb-modified. These results have been integrated into the MS (Lines 399-402, Supplementary Table 13).

- The Discussion does not address the critical point of how hydroxybutyrylation could affect gene expression. Can hydroxybutyrylation affect histone H3 affinity for DNA, as acetylation does?

Response:

Thanks for the suggestion. Previous studies have suggested a potential link between histone Kbhb and chromatin accessibility (Terranova et al., 2021; Zhou et al., 2022). As histone Kbhb is considered a positive mark, it is hypothesized that, similar to acetylation, Kbhb may neutralize the positive charge on histones, thereby weakening histone–DNA interactions and leading to a more open chromatin state that promotes gene expression. However, the direct relationship between histone Kbhb and chromatin accessibility has not been extensively investigated. To address this, we performed ATAC-seq analysis on BHB-treated and control (CK) samples, which revealed that elevated Kbhb levels are associated with increased chromatin accessibility (Supplementary Fig. 13), supporting its role in chromatin remodeling.

We have incorporated this into the Discussion section (Lines 432-438).

Reference

Terranova, C. J. et al. Reprogramming of H3K9bhb at regulatory elements is a key feature of fasting in the small intestine. *Cell Rep.* 37(8), 110044 (2021).

Zhou, T. et al. Function and mechanism of histone β -hydroxybutyrylation in health and disease. *Front Immunol.* 13, 981285 (2022).

- Also, is H3K9bhb expected to affect transcription through chromatin accessibility? If so, this could be tested using ATAC-seq or MNase-seq analyses upon BHB application.

Response:

To test whether BHB application affects chromatin accessibility, we performed ATAC-seq on both control (CK) and BHB-treated samples. The data analysis revealed a significant increase in chromatin accessibility upon BHB treatment (Supplementary Fig. 13b). Additionally, genes with increased H3K9bhb deposition showed higher chromatin accessibility in BHB-treated plants compared to CK (Supplementary Fig. 13c). This suggests that H3K9bhb regulates transcription by influencing chromatin accessibility. These results have been incorporated into the MS and discussed in the Discussion section (Lines 249-257, Lines 439-443, Supplementary Fig. 13).

Reviewer #3 (Remarks to the Author):

Lysine β -hydroxybutyrylation (Kbhb) is a new type of histone mark, while its prevalence and function remain unclear in plants. Xu et al. performed a genome-wide profiling of histone H3K9bhb in rice, and found histone H3K9bhb marks were highly enriched at transcription start sites (TSSs), which positively correlated with the active histone marks H3K4ac, H3K9ac, and H3K4me3. Further ChIP-seq data showed that rice H3K9bhb modification was enriched at the genes involved in the defense response to fungal infection. Increasing H3K9bhb modification in rice, results in higher resistant on *U. virens* infection, suggesting H3K9bhb may have a key role in plant defense mechanisms. The authors also found that SRT2 and HDA705 had notable de-Kbhb activity in rice. At last, they also performed a proteome-wide identification of Kbhb substrates in rice flowers, and found Kbhb modified proteins play important roles in plant metabolism.

The overall story is interesting, which includes huge amount of data and disclosed the distribution, deposition, and function of histone Kbhb in rice. However, this manuscript spends most of the space to focus on the omics data, failed to reveal the fundamental mechanisms of Kbhb modification in rice. Although there is evidence that Kbhb help to increase fungal infection in rice, whereas it lacks the direct evidence to support the function of Kbhb modification in rice immune system. HDACs have de-Kbhb activity, while there is no evidence that showed SRT2 and HDA705 have roles in response to fungal infection. To prove the direct writer and eraser of histone Kbhb in rice, it needs more evidence that should include in vivo, in vitro and genetic experiments, and the current evidence is not sufficient. The last part “Proteome-wide identification of Kbhb substrates in rice flowers” seems to stand alone and does not effectively contribute to the main points of the paper. It appears that these data points are not well - integrated into the overall argument, and as a result, they do not enhance the validity or persuasiveness of your research.

All in one, the current manuscript is not suitable for publication in Nature Communications, however, it is worth a re-submit opportunity to take a step back and carefully assess the overall logic and essential evidence showed in this manuscript.

Response:

Thanks for the valuable suggestion.

Application of exogenous BHB significantly enhanced disease resistance in rice plants (Fig. 4). This treatment elicited a series of immune responses, including reactive oxygen species (ROS) burst,

callose deposition, activation of MAPK signaling cascades, and transcriptional induction of defense-related genes (Fig. 4). Concurrently, BHB treatment led to elevated levels of H3K9bhb modification in plant tissues (Fig. 5a). Through functional analyses, we identified the deacetylases SRT2 and HDA705 as negative regulators of rice immunity, with HDA705 specifically modulating H3K9bhb deposition to repress the expression of immunity-associated genes (Fig. 7, Supplementary Fig. 21 and 22). These results collectively demonstrate that H3K9bhb serves as a biologically relevant histone modification that contributes to the regulation of immune responses in rice.

In this revision, we further validated the role of HDA705 by generating a point mutation (H154A) in its catalytic domain, as reported in our previous study (Xu et al., 2025), and confirmed that this mutation significantly reduced its de-H3K9bhb activity (Supplementary Fig.17). We also examined H3K9bhb levels in *HDA705*-overexpression lines and conducted ChIP-qPCR analyses in both *hda705* mutants and overexpression lines to demonstrate HDA705's direct role in regulating H3K9bhb enrichment at selected immunity-related gene loci (Figure 7, Supplementary Fig.17 and 21). These results strengthen the evidence establishing HDA705 as a functional eraser of H3K9bhb in rice.

In addition, we performed H3K9bhb ChIP-seq in the *srt2* mutant background, which further confirmed the de-H3K9bhb activity of SRT2 (Supplementary Fig.24). To assess the functional relevance, we examined the disease resistance of both *srt2* and *hda705* mutants against *Magnaporthe oryzae*, and observed increased resistance in both mutants (Supplementary Fig.18). To test whether enhanced resistance was due to altered H3K9ac levels, we conducted H3K9ac ChIP-seq in the *hda705* background and found that HDA705-regulated H3K9ac target genes are primarily involved in metabolic processes and development (Supplementary Fig. 23), functionally distinct from H3K9bhb-regulated genes.

To improve the logical flow of the manuscript, we have reordered the figures as suggested by Reviewer 4, moving the previous Fig. 3 ahead of Fig. 2. Additionally, we performed ChIP-re-ChIP assays to confirm the mutual exclusivity of H3K9bhb and H3K9ac modifications at the same loci (Supplementary Fig.7). We also added H3K9ac ChIP-qPCR results for selected immunity-related genes in both infected (IF) and control (CK) rice spikelets (Supplementary Fig.15), further supporting the regulatory role of H3K9bhb in transcription regulation of immunity genes. Moreover, to investigate the impact of H3K9bhb on chromatin accessibility, we generated and analyzed ATAC-seq data, providing further insights into the epigenomic landscape shaped by this histone modification (Supplementary Fig.13).

Regarding the final part of the manuscript, "Proteome-wide identification of K9bhb substrates in rice flowers", while we believe the dataset provides valuable information for the field, we acknowledge the reviewer's concern about its integration into the main narrative. Therefore, we have moved this section (previously Figure 8) to the Supplementary Data (Supplementary Fig. 25) to de-emphasize its role in the MS.

We hope that these revisions and the new evidence provided have sufficiently addressed the reviewer's concerns and significantly improved the manuscript.

Reference

Xu, Q. et al. Histone H4K8hib modification promotes gene expression and regulates rice immunity, *Mol. Plant* 18, 9–13 (2025).

Reviewer #4 (Remarks to the Author):

Qiutao et al. investigate the presence of H3K9bhb as an epigenetic mark in rice and its role in plant defenses. They discovered that H3K9bhb is associated with actively transcribed genes and identified functional differences between this mark and H3K9ac. Interestingly, H3K9bhb is enriched in genes associated with plant defenses. They also demonstrate that exogenous BHB application induces H3K9bhb accumulation and enhances plant defenses, while plant infection triggers endogenous accumulation of this mark in defense genes. Furthermore, they reveal that OsSRT2 and OsHDA705 can remove this epigenetic mark.

I found the manuscript highly interesting and easy to read, providing valuable insights for further research. However, I have a major concern that needs to be addressed. The authors present enough evidence of the existence and/or presence of H3K9bhb in rice, but an assay demonstrating the *in vivo* specificity of the antibody would be better. While Figure S1 shows the antibody's specificity *in vitro*, comparing its cross-reactivity with H3K14bhb, H3K9cr, H3K9la, and H3K9ac, I remain concerned about potential cross-reactivity with other proteins, particularly those in the nucleus, where 15% of the proteins immunoprecipitated with anti-Kbhb are located (Figure 8). This suggests that other proteins beyond those tested in Figure S1 could also be targeted in ChIP assays or western blots, as shown in Figure 5.

Response:

Thanks for the suggestion. To further test the specificity of the anti-H3K9bhb antibody, we selected two nuclear-localized proteins that were identified as Kbhb-modified in our Kbhb proteome analysis (Supplementary Fig. 3a, Supplementary Table 12), with histone H3 included as a positive control. We first extracted the nuclei from rice plants and then performed immunoprecipitation using the anti-H3K9bhb antibody. Afterward, we checked for the presence of the respective proteins. The results showed that only H3 was detected, while RPS6 and HSP70 were not detected (Supplementary Fig. 3b). These results further confirm the *in vivo* specificity of the anti-H3K9bhb antibody.

I do not question the broad conclusions, but without proper endogenous controls, the presence of this epigenetic mark in the genome could be overestimated or biased (especially since many of the IPed proteins related to plant-pathogen interactions, as shown in Figure 8f, could skew the results). Could the authors perform a WB with anti-H3K9bhb, followed by LC-MS/MS of the purified band corresponding to H3K9bhb, probably in BHB-treated and untreated plants? I would expect an enrichment of H3K9 peptides with bhb PTM rather than other PTMs or unrelated proteins. Alternatively, overexpressing HDA705 or SRT2 to reduce H3K9bhb levels and use ChIP-qPCR to assess H3K9bhb presence on specific target genes? While this approach might be biased if these HDACs also de-Kbhb other cross-reacting proteins, it would help address this concern.

Response:

Thanks for the suggestion. To further verify the specificity of the anti-H3K9bhb antibody, we conducted immunoprecipitation assays using two nuclear-localized proteins (See previous response). As suggested by the reviewer, we also examined H3K9bhb levels in *HDA705*-overexpressing plants and found that overexpression of *HDA705* led to a significant reduction in H3K9bhb levels in rice cells (Supplementary Fig. 17). Furthermore, ChIP-qPCR analysis confirmed that genes exhibiting increased H3K9bhb levels in *hda705* mutants (Figure 7g) showed a significant decrease in H3K9bhb enrichment in *HDA705*-overexpressing plants (Supplementary Fig. 21). These results provide additional evidence supporting the specificity of the anti-H3K9bhb antibody and the regulatory role of *HDA705* in modulating H3K9bhb levels.

Minor Concerns:

Could the authors clarify Figure 2C? Why does the peak of reads (from H3K9bhb ChIP-seq) appear near the TTS, unlike Figures 1G and 2A, where peaks are mainly near the TSS?

Response:

Sorry for the typo. It refers to the TSS in Figure 3c (Fig. 2c has been moved to Fig. 3c in this revision). Thanks for pointing this out. We have corrected it.

The authors present three clusters of H3K9bhb-marked genes alongside other epigenetic marks and analyze GO terms for cluster 1, showing an enrichment of stress-induced genes among others. What about the genes in the other two clusters? Do they differ from those in cluster 1? This could hint at distinct roles for H3K9bhb in combination with different epigenetic marks.

Response:

Thanks for the suggestion. We have performed GO pathway analysis for genes in clusters 2 and 3. The results showed that genes in cluster 2 were most enriched in processes such as peptidyl-threonine phosphorylation, G2/M transition of the mitotic cell cycle, regulation of circadian rhythm, retrotransposition, and transposition. In contrast, cluster 3 was most enriched in processes related to protein transport, RNA splicing, ncRNA metabolic processes, microtubule-based processes, carbohydrate derivative biosynthesis, DNA replication, plastid organization, and cell development. These results from clusters 2 and 3 differ significantly from those in cluster 1, suggesting distinct roles for H3K9bhb in combination with different epigenetic marks. These findings have been integrated into the MS (Lines 189-192, Supplementary Fig. 8).

I suggest presenting the results in Figure 3 before those in Figure 2. Figure 3 provides a broader context (e.g., GO analysis and correlation with H3K9ac), while Figure 2 distinguishes between different gene classes marked by H3K9bhb. This is a suggestion for the authors to consider.

Response:

Thank you for the suggestion. We have reordered the figures, presenting Figure 3 before Figure 2, and have also adjusted the corresponding results section in the manuscript accordingly

The authors show that *SRT2* and *HDA705* regulate H3K9bhb levels in pathogen-responsive genes. The ChIP-seq analysis in *hda705* mutant plants and the *in vitro* de-Kbhb activity assay suggest that *HDA705* may have de-Kbhb activity. To strengthen this point, I suggest immunoprecipitating *HDA705* to test its association with H3K9bhb-enriched sequences *in vivo* via ChIP-qPCR on specific targets.

Response:

Thank you for the suggestion. We have examined HDA705 binding to its target genes using *HDA705*-overexpression plants (with a 3 × FLAG tag fused at the N-terminus) and performed ChIP-qPCR with an anti-FLAG antibody. The results showed that genes with decreased H3K9bb exhibited higher HDA705 binding compared to the control. These findings have been incorporated into the Supplementary Fig. 21.

The statistical analysis in Figure 3C is missing.

Response:

Thanks for the suggestion. We have added the statistical analysis in Figure 2c (Figure 3c has been moved to Figure 2c in this revision).

Line 116: "genes (d)" should be corrected to "genes (Fig. 1d)."

Response:

Thanks for the suggestion. We have corrected it (Line130).

Reviewer's Comments:

Reviewer #1 (Remarks to the Author):

Authors have resolved all my concerns, it can be accepted now.

We sincerely thank the Reviewer #1 for the positive comments on our revision.

Reviewer #2 (Remarks to the Author):

The revised version of the manuscript incorporates a significant amount of additional workload, including the requested analyses of H3K9ac/H3K9bhb relationships through sequential ChIP (Re-ChIP), and the apparent influence of H3K9 hydroxybutyrylation on gene expression by modulating chromatin accessibility. In my opinion the manuscript is suitable for publication in Nature Coms provided that:

We sincerely thank Reviewer #2 for the valuable and constructive comments, which have significantly improved our manuscript.

1) the conclusion that H3K9ac and H3K9bhb are two mutually exclusive histone H3K9 modifications is better argued in the Discussion by compiling all relevant evidence from this study - not just Re-ChIP-qPCR analysis of a few selected genes.

Thanks for your suggestion. We have revised the Discussion section (Lines 458 - 462) by integrating the evidence presented in the manuscript to support the conclusion that H3K9ac and H3K9bhb are two mutually exclusive histone H3K9 modifications.

2) this conclusion is used in all instances for accurate data interpretation, hence proposals that genes are dually marked by H3K9ac and H3K9bhb (e.g. line 144) should be re-visited or more cautiously formulated. As discussed, apparent dual marking of a given gene could result from individual marking by each of these two modifications in different cells.

Thanks for your suggestion. We have revised the statements regarding genes dually marked by H3K9ac and H3K9bhb at line 144 (line 144-151 in the revised version) and elsewhere (line 165-166), emphasizing the potential influence of cell-to-cell heterogeneity.

3) the ATAC-seq data are better integrated with gene expression (RNA-seq) and chromatin profiling (H3K9bhb ChIP-seq) to sustain the possibility that H3K9bhb is linked to increased DNA accessibility and gene expression control at specific immunity gene targets.

Thanks for your suggestion. As suggested, we have performed integrative analyses combining gene expression (RNA-seq), chromatin profiling (H3K9bhb ChIP-seq), and chromatin accessibility (ATAC-seq) data. The results support the association of H3K9bhb with enhanced DNA accessibility and transcriptional activation at specific immunity-related gene loci. These findings have been incorporated into the manuscript (Lines 263-265; Supplementary Fig. 13d, Supplementary Data 5).

Reviewer #3 (Remarks to the Author):

The current version is much better, while there are still some concerns about logic problems in this manuscript. If the authors could fully address the two questions as described above, I think the manuscript would be ready for acceptance by Nature Communications.

1. For the second part “Functional divergence of H3K9bhb and H3K9ac” (Line 142), why the authors want to investigate the correlation between H3K9bhb and H3K9ac? Are there any evident to show that H3K9bhb functions together with H3K9ac? There should have the transitional phrases to explain why the authors want to start the study of this part. Similar problems are shown in the third part “Comparative analysis of H3K9bhb, histone H3 acetylation, and histone H3 methylation modification” (Line 172).

Thanks for the suggestion.

For "Functional divergence of H3K9bhb and H3K9ac" part: “We have the following sentence to have the transitional phrases and explain why the we start the study of this part: “Given that both H3K9bhb and H3K9ac occur at the same lysine residue and are associated with transcriptional activation, it is important to assess whether they function redundantly or exhibit distinct regulatory roles in gene expression (Line 144-146). ”

For "Comparative analysis of H3K9bhb, histone H3 acetylation, and histone H3 methylation" part: We have revised the first sentence of this part to have the transitional phrases and explain why the we start the study of this part: “To further explore the epigenetic landscape shaped by H3K9bhb, we expanded our analysis to include well-characterized histone modifications such as H3 acetylation and methylation, aiming to uncover how these marks coordinate or compete in regulating gene expression (Line 178-181).”

2. Are there any rice mutants with null or low level of β -hydroxybutyrylation? If lower down the total level of β -hydroxybutyrylation in vivo, what are the development or disease resistant phenotypes of these mutants?

Thank you for your insightful question. To the best of our knowledge, no rice mutants with null or significantly reduced levels of β -hydroxybutyrylation (Kbhb) have been reported to date. Consequently, the developmental or disease resistance phenotypes associated with reduced Kbhb levels cannot be investigated at this stage. Nevertheless, this represents a promising direction for future research.

Reviewer #4 (Remarks to the Author):

The authors have included sufficient results to address my concerns. In addition, the revisions made in response to the other reviewers’ comments have further improved the manuscript. Overall, I find the work very interesting and valuable.

We sincerely thank Reviewer #4 for the positive evaluation and encouraging comments.